# Online Learning for Obstacle Avoidance

David Snyder[†,1,3]    Meghan Booker[1]    Nathaniel Simon[1]

Wenhan Xia[2,3]    Daniel Suo[2,3]    Elad Hazan[2,3]    Anirudha Majumdar[1,3]

**Abstract:** We approach the fundamental problem of obstacle avoidance for robotic systems via the lens of online learning. In contrast to prior work that either assumes worst-case realizations of uncertainty in the environment or a stationary stochastic model of uncertainty, we propose a method that is efficient to implement and provably grants *instance-optimality* with respect to perturbations of trajectories generated from an open-loop planner (in the sense of minimizing worst-case *regret*). The resulting policy adapts online to realizations of uncertainty and provably compares well with the best obstacle avoidance policy *in hindsight* from a rich class of policies. The method is validated in simulation on a dynamical system environment and compared to baseline open-loop planning and robust Hamilton-Jacobi reachability techniques. Further, it is implemented on a hardware example where a quadruped robot traverses a dense obstacle field and encounters input disturbances due to time delays, model uncertainty, and dynamics nonlinearities.

**Keywords:** Regret Minimization, Obstacle Avoidance, Online Learning

## 1   Introduction

The problem of obstacle avoidance in motion planning is a fundamental and challenging task at the core of robotics and robot safety. Successfully solving the problem requires dealing with environments that are inherently uncertain and noisy: a robot must take into account uncertainty — external disturbances and unmodeled effects, for example — in its own dynamics and those of other agents in the environment. Approaches for tackling the obstacle avoidance problem in robotics typically fall under two categories: (i) methods that attempt to construct stochastic models of uncertainty in the agents' dynamics and use the resulting probabilistic models for planning or policy learning, and (ii) methods that construct plans that take into account worst-case behavior. In Sec. 2 we give a more detailed overview of both classes of approaches.

In this paper, we are motivated by Vapnik's principle: *"when solving a given problem, try to avoid solving a harder problem as an intermediate step."* Constructing accurate models of disturbances and agent dynamics is perhaps more complicated than the task of obstacle avoidance in motion planning, as practical uncertainties rarely conform to assumptions made by the two classes of approaches highlighted above. As an example, consider a quadruped robot navigating through an (*a priori* unknown) obstacle field subject to unmodeled dynamics and external disturbances in the input channel (Fig. 1). Constructing accurate probabilistic models of disturbance and obstacle variations is challenging, and may cause the robot to violate safety in out-of-distribution settings. In contrast, worst-case assumptions may lead to extremely conservative behavior. This motivates the need for *online learning* methods that adapt to the *particular context* encountered by the robot.

**Statement of Contributions.**   In this work, we pose the problem of obstacle avoidance in a *regret minimization* framework and build on techniques from non-stochastic control. Our primary contribution is a trust-region-based online learning algorithm for the task of obstacle avoidance, coupled with provable regret bounds that show our obstacle avoidance policy to be comparable to the *best policy in*

---

[†]Corresponding Author: dasnyder@princeton.edu.  [1]Intelligent Robot Motion (IRoM) Lab, Princeton University. [2]Department of Computer Science, Princeton University. [3]Google DeepMind.

7th Conference on Robot Learning (CoRL 2023), Atlanta, USA.

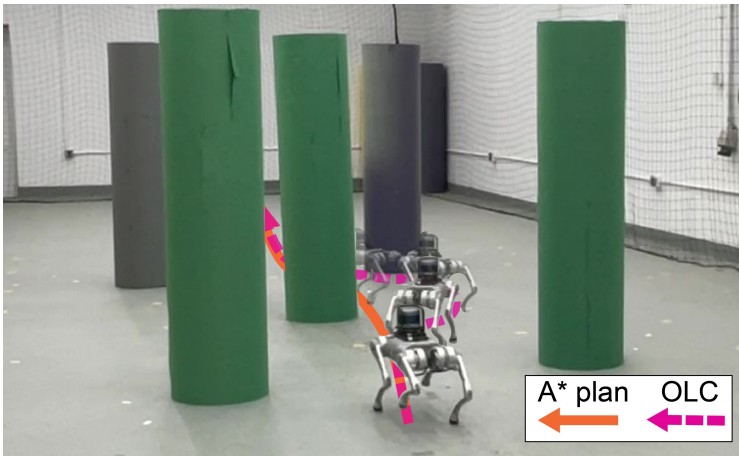

Figure 1: A quadruped robot is tasked with traversing a course with densely placed obstacles to a goal. During traversal, the robot is subject to sensor noise, time delays, input-channel disturbances, and nonlinearities in the closed-loop dynamics, which can render optimistic nominal plans (orange) unsafe. Our online learning control (OLC) algorithm (dashed pink) corrects for this to give a wider margin of error when moving through the course.

*hindsight* from a given class of closed-loop policies. This type of theoretical performance guarantee is nonstandard, and allows us to flexibly adapt to the behavior of the uncertainty in any *instance* of the obstacle avoidance problem without making *a priori* assumptions about whether the uncertainty is stochastic or adversarial. Further, the resulting method is computationally efficient. The method is applied to dense obstacle environments with complex unmodeled dynamics, and demonstrates improved performance where open-loop planners and overly-robust methods can respectively struggle. We additionally show the efficacy of our method with hardware demonstrations where a quadruped robot has to navigate dense obstacle fields subject to time delays and input-channel disturbances.

## 2   Related Work

Effective motion planning is a central challenge within robotics of continuing and significant interest [1, 2, 3]. While planning is well-developed in deterministic settings, robustness remains a major challenge in the presence of unmodeled uncertainty. Existing techniques typically fall into one of two categories: (i) methods that make assumptions on the distribution of uncertainty, or (ii) methods that assume worst-case disturbances. Below, we set our work within this context and discuss techniques from online learning; we will leverage the latter to develop our framework for obstacle avoidance.

**Planning under uncertainty.**   A popular approach to account for uncertainty in motion planning is to assume knowledge of the uncertainty distribution. One early method in this vein utilizes *chance constraints* to bound the probability of collision under stochastic uncertainty [4] and has been subsequently extended to encompass many sources of stochastic uncertainty in robotics [5] - [8]. Further development has utilized partially observable Markov decision processes (POMDPs) to account for state uncertainty [9] - [12]. These approaches are able to provide strong guarantees on safety, albeit under generally restrictive assumptions on the uncertainty distribution (e.g., i.i.d. Gaussian uncertainty); our approach does not assume knowledge of the distribution of uncertainty, yet provides regret bounds even in the presence of non-Gaussian and non-stationary noise.

Recently, learning-based planning techniques relying on *domain randomization* have demonstrated significant empirical success [13] - [19] by specifying a distribution of uncertainty over various simulation parameters to train robust policies. Combining this domain randomization with online identification of uncertain parameters has been proposed [20, 21]; however, despite these methods' empirical successes, they still rely on real-world environments being well-represented by the distribution of uncertainty used in (relatively extensive) training. By contrast, our approach focuses on settings where it may be challenging to specify the uncertainty, and we do not require expensive training of policies in simulation to provide theoretical guarantees in the form of bounded regret.

**Reachability-based robust planning.** Hamilton-Jacobi (HJ) Reachability-based [22, 23] and trajectory library-based [24, 25] robust planning techniques assume *worst-case* realization of uncertainty. As such, they provide adversarial certificates of safety for path planning problems via the construction of representations (or outer approximations) of the safe and unsafe regions of the state space conditional on the robot dynamics, obstacle placement, and disturbance size. This formalism is similar to related results in adaptive control [26, 27], which prove stability in the face of disturbances for specific planning controllers, even in the presence of certain non-convex obstacles. HJ methods, which generally suffer from the curse of dimensionality [28] (despite the existence of speed-ups in certain settings [29]), use the formalism of the differential game [30] to provide a "global" notion of safe and unsafe sets [31]. In comparison, robust trajectory libraries, which are usually computed using convex programs [32], provide safety guarantees in the form of robust "tubes" or "funnels" [33, 25] that encompass only the nominal trajectory (or hypothesized trajectories) within the space.

Our work differs from these methods via guaranteeing "instance-optimality;" namely, the regret minimization framework allows us to adapt to the *specific* nature of observed disturbances. Our method is effective in *both* stochastic and adversarial regimes, we do not sacrifice too much performance in "benign" environments to provide guaranteed robust performance in more adversarial cases.

**Online learning for control.** Our work makes significant use of *online learning* [34] to make guarantees on *regret*, which is the difference between the algorithm's performance and that of the *best policy in hindsight* (once disturbances are realized) from a given class of closed-loop policies. Several canonical control-theoretic results have recently been cast as problems in online learning [35, 36], providing interesting generalizations to established control results like the linear-quadratic regulator [37, 38, 39] and H$_\infty$ robust control [40]. Results in optimal sample complexity [41] and synthesis for unknown linear systems [42] illustrate further generalizations of standard control theory.

Standard control formulations are efficiently solvable due to a convex objective. However, "higher-level" decision-making tasks like obstacle avoidance often have non-convex objective functions (e.g., maximizing the distance to the nearest obstacle). Fortunately, some non-convex objectives admit "hidden convexity" — that they can be reformulated (via transformations, relaxations, or conversions to a dual formulation — see [43] for a survey) into equivalent optimization problems that *are convex*. This allows for efficient solutions (e.g., [44, 45, 46]) to problems that nominally would be hard to solve (e.g., [47]). Our work gives such a formulation for the task of obstacle avoidance.

## 3   Problem Formulation and Preliminaries

Consider a discrete-time dynamical system with state $\bar{\mathbf{x}}$ and control input $\bar{\mathbf{u}}$. A planning oracle $\mathcal{O}_\mathcal{T}(\bar{\mathbf{x}}_0)$ takes in an initial state and generates a nominal state trajectory with associated control inputs $\mathcal{T} = \{\bar{\mathbf{x}}_t^0, \bar{\mathbf{u}}_{t-1}^0\}_{t=1}^T$. Here, $\mathbf{x} \in \mathbb{R}^{d_x}$ and $\mathbf{u} \in \mathbb{R}^{d_u}$. We design a robust obstacle avoidance controller that will update the trajectory online to avoid local obstacles. Intuitively, this is a faster "safety inner loop" for the slower trajectory planning stage $\mathcal{O}_\mathcal{T}$, keeping the agent safe from external features. For analytical tractability, we assume that the dynamics of perturbations of the nominal trajectory are discrete-time linear. Defining $\mathbf{x}_t = \bar{\mathbf{x}}_t - \bar{\mathbf{x}}_t^0$ and $\mathbf{u}_t = \bar{\mathbf{u}}_t - \bar{\mathbf{u}}_t^0$, this assumption becomes

$$\mathbf{x}_{t+1} = A_0\mathbf{x}_t + B\mathbf{u}_t + \mathbf{w}_t, \tag{1}$$

where $\mathbf{w}_t \in \mathbb{R}^{d_w}$ is a bounded, unknown, possibly-adversarial disturbance[1]. For many practical systems the linear dynamics assumption is reasonable; one example is a control-affine system with feedback linearization. Additionally, $\mathbf{w}_t$ can encompass small, unmodeled nonlinearities. Our task is to construct this "residual" controller generating $\mathbf{u}_t$ to avoid obstacles. As such, $\mathcal{O}_\mathcal{T}$ is the optimistic, goal-oriented planner (in practice, an off-the-shelf algorithm, e.g., [48]) and our controller is the safety mechanism that becomes active *only when needed* and in a provably effective manner.

---

[1]The results here are presented for linear time-invariant (LTI) systems. Additionally, we present the case $d_w = d_x$, which affords the disturbances the most 'power'; these results immediately extend to the inclusion of a non-identity disturbance-to-state matrix and $d_w \le d_x$ in Eqn. 1.

## 3.1 Safety Controller Objective

A controller trying to avoid obstacles needs to maximize the distance to the nearest obstacle, subject to regularization of state deviations and control usage. Assume a sensor mechanism that reports all "relevant" obstacle positions (e.g., within a given radius of the agent). The optimization problem, for a trajectory of length $T$, safety horizon of length $L \leq T$, and obstacle positions $\mathbf{p}_t^j$ denoting the j$^{th}$ sensed obstacle at time $t$, is

$$\max_{A \in \mathcal{A}} C_{\text{obs}}(A), \quad \text{where:} \tag{2}$$

$$C_{\text{obs}}(A) := \sum_{t=1}^{T} \min_{\tau \in [L]} \min_{j} \|\mathbf{x}_{t+\tau}^A - \mathbf{p}_t^j\|_2^2 - \|\mathbf{x}_t^A\|_Q^2 - \|\mathbf{u}_t^A\|_R^2.$$

Here, $\mathcal{A}$ is the set of online algorithms that choose actions for the controller, and $\mathbf{x}_t^A$ denotes the realized state trajectory conditioned on actions $\mathbf{u}_t^A \sim A \in \mathcal{A}$. The last two terms represent quadratic state and action costs; these costs[2] are very common objectives in the control literature, but serve primarily here to regularize the solution to the obstacle avoidance task. Note that, though the collision avoidance objective is relaxed, it remains a nonconvex quadratic penalty term rendered additionally complex due to the discrete selection over time and obstacle indices of the minimal-distance obstacle in the first term of $C_{obs}$. From here, we model the optimal policy search in the online control paradigm [38]. This allows us to define the regret-based safety metric with respect to the best achievable performance in hindsight. For a sufficiently powerful policy comparator class, we achieve meaningful guarantees on the safety of the resulting controller.

## 3.2 Regret Framework for Obstacle Avoidance

Leveraging the class of linear dynamic controllers (LDCs) as comparators [40, 38], we use a disturbance-action controller design, which combines state feedback with residual obstacle-avoidance:

$$\mathbf{u}_t = K\mathbf{x}_t + \mathbf{b}_t + \sum_{i=1}^{H} M_t^{[i]} \mathbf{w}_{t-i}. \tag{3}$$

Here, $[i]$ indexes the history length and $t$ denotes the time index of the decision. We rearrange the expression by moving the state feedback out of $\mathbf{u}_t$, defining $\tilde{\mathbf{u}}_t = \mathbf{u}_t - K\mathbf{x}_t$ and $\tilde{A} = A_0 + BK$. Then the system dynamics (Eqn. 1) are equivalently

$$\mathbf{x}_{t+1} = \tilde{A}\mathbf{x}_t + B\tilde{\mathbf{u}}_t + \mathbf{w}_t. \tag{4}$$

The comparator class $\Pi$ will be the class of LDCs parameterized by $M$; this class has provably expressive approximation characteristics [38]. The regret is defined using quantities from Eqn. 2:

$$\text{Reg}_T(A) = \sup_{w_1, \ldots, w_T} \left\{ \max_{M \in \Pi} C_{\text{obs}}(M) - C_{\text{obs}}(A) \right\}. \tag{5}$$

A sublinear bound on Eqn. 5 implies that the adaptive sequence $M_t$ selected by low-regret algorithm $A$ will perform nearly as well as the *best fixed policy $M^*$ in hindsight, for all realizations of uncertainty* within the system. Thus, sublinear regret bounds establish finite-time (near) optimality; the policy performs nearly as well as an optimal policy that has *a priori* knowledge of the realized disturbances.

## 3.3 Trust Region Optimization

To provide guarantees on regret, the method presented in Sec. 4.2 will construct sequential *trust region optimization* instances. A trust region optimization problem [49] is a nominally non-convex optimization problem that admits "hidden convexity". One can reformulate a trust region instance (see Def. 1) via a convex relaxation in order to solve it efficiently [44, 50].

---

[2]Here, we omit the fully general LQR costs (time-varying $Q_t$ and $R_t$) for simplicity of presentation; however, the results we show will *also* hold for this more general setting.

**Definition 1** (Trust Region Instance). *A trust region instance is defined by a tuple $(P, \mathbf{p}, D)$ with $P \in \mathbb{R}^{d \times d}$, $\mathbf{p} \in \mathbb{R}^d$, and $D > 0$ as the optimization problem*

$$\max_{\|\mathbf{z}\| \le D} \left\{ \mathbf{z}^T P \mathbf{z} + \mathbf{p}^T \mathbf{z} \right\}. \tag{6}$$

Throughout the remainder of this paper, we will use "trust region instance" to refer interchangeably to instances of Def. 1 and the implicit, equivalent convex relaxation.

# 4 Methodology, Algorithm, and Regret Bound

## 4.1 Intuitive Decomposition of the OLC Algorithm Control Signal

The intuition of our control scheme is to allow for online, optimization-based feedback to correct for two key sources of failure in obstacle avoidance: (1) non-robust (risky) nominal plans, and (2) external disturbances. The former can be thought of as errors in planning – that is, they arise *when the nominal path is followed exactly.* Paths that move very close to obstacles or that pass through them (e.g., due to sensor noise) would be examples of this problem. The presence of the bias term in the OLC framework accounts for control input to correct these errors. The latter challenge concerns *deviation from nominal trajectories* caused by errors in modeling, time discretization, and physical disturbances. The linear feedback term in the OLC framework accounts for these errors.

## 4.2 Theoretical Method and Contributions

We utilize the "Follow the Perturbed Leader" (FPL) method [35] in tandem with an optimization oracle [51, 52] from the online convex optimization [53] literature. In this area, [40] develops a method to generate adversarial disturbances for linear systems online, and extends the FPL algorithm to the setting with memory. Further, [54] gives regret bounds for many game scenarios in which the players have varying cost landscapes. Several contributions of this paper lie in *reductions*: we show that optimizing Eqn. 8 is equivalent to finding the equilibrium solution of a general-sum two-player game for which every control-player instance is a trust-region problem, and that both players have low-regret algorithms to solve for the optimal policies. This allows us to use the results in [54]. Additionally, we show that extension to the multi-step setting in Eqn. 9 remains a trust region instance in a 'lifted' game, allowing us to use the FPL results ( [52, 51, 40]).

## 4.3 Algorithm Exposition and Regret Bound

We formulate the obstacle avoidance controller as an online non-convex FPL algorithm in Alg. 1. At time $t$, the agent generates a control input $\tilde{\mathbf{u}}_t$ via Eqn. 3 by playing $M_t^{[1:H]} \in \{\tilde{M}\}$. The state dynamics are then propagated to reveal the new state $\mathbf{x}_{t+1}$, and the realized $\mathbf{w}_t$ is reconstructed in hindsight from the state. The robot simultaneously uses its sensors to observe local obstacle positions.

The key conceptual step is then the following: given past reconstructed disturbances and obstacle locations, one can construct a instantaneous reward function (Eqn. 8) at time $t + 1$. This is a function of a *counterfactual* set of gains $\{\tilde{M}\}$, where $\mathbf{x}_{t+1}^{\tilde{M}}$ and $\tilde{\mathbf{u}}_t^{\tilde{M}}$ correspond to the state and control input that *would have resulted* from applying $\tilde{M}$ given the realized disturbances, obstacle locations, and the initial state $\mathbf{x}_0$. The key algorithmic step is to solve for the optimum $M_{t+1}^{[1:H]} \in \{\tilde{M}\}$ in Eqn. 9, which is the Follow-the-Perturbed-Leader component of the algorithm ($\tilde{M} \bullet P_0$ is the perturbation). The resulting sublinear regret bound is given in Thm. 2.

**Theorem 2** (Regret Bound for Online Obstacle Avoidance). *Consider an instance of Alg. 1, with Alg. 2 (see Supp. B) acting as a subroutine optimizing Eqn. 9. Then the regret attained by Memory FPL will be*

$$\tilde{\mathcal{O}}(poly(\mathcal{L})\sqrt{T}), \tag{7}$$

*where $\mathcal{L}$ is a measure of the instance complexity.*

**Algorithm 1** Online Learning Control (OLC) for Obstacle Avoidance

---

**Input**: Observed obstacle positions $\{\mathbf{p}_0^j\}_{j=1}^K$, history length $H = \mathcal{O}(\log T)$.
**Input**: Full horizon $T$, algorithm parameters $\{\eta, \epsilon, \lambda\}$, initial state $\mathbf{x}_0$.
**Input**: Open-loop plan: $\bar{\mathbf{u}}_t^o$ for $t = 1, ..., T$.
**Initialize**: Closed-loop correction $M_0^{[1:H]}$, fixed perturbation $P_0 \sim \text{Exp}(\eta)^{d_u \times H d_w}$.
**Initialize**: Play randomly for $t = \{0, ..., H-1\}$, observe rewards, states, noises, and obstacles.
**for** $t = H$ **until** $T - 1$ **do**
    Play $M_t^{[1:H]}$, and observe state $\mathbf{x}_{t+1}$ and obstacles $\{\mathbf{p}_{t+1}^j\}_{j \in [k]_{t+1}}$.
    Reconstruct disturbance $\mathbf{w}_t$ using observed $\mathbf{x}_{t+1}$.
    Construct the reward function in $\tilde{M}$:

$$\ell_{t+1}(\tilde{M}) = \min_{j \in [k]} \left\{ \|\mathbf{x}_{t+1}^{\tilde{M}} - \mathbf{p}_{t+1}^j\|_2^2 \right\} - \|\mathbf{x}_{t+1}^{\tilde{M}}\|_Q^2 - \|\tilde{\mathbf{u}}_t^{\tilde{M}}\|_R^2. \tag{8}$$

    Receive realized reward $l_{t+1}(M_t^{[1:H]})$.
    Solve for 'perturbed leading policy' $M_{t+1}^{[1:H]}$ as the solution to:

$$\underset{\|\tilde{M}\| \leq D_M}{\arg\max} \left\{ \sum_{\tau=1}^{t+1} \ell_\tau(\tilde{M}) + \lambda(\tilde{M} \bullet P_0) \right\}. \tag{9}$$

**end for**

---

## 5 Experiments

We demonstrate the effectiveness of our method in simulated and physical environments. The simulated environment in Sec. 5.1 considers a 2D racing problem. The hardware experiments in Sec. 5.2 are conducted on a Go1 quadruped robot [55] avoiding obstacles in the presence of unmodeled nonlinear dynamics, sensor noise, and time delays, as shown in Fig. 1. In both settings, we demonstrate the benefits of our approach as compared to baselines that use, resp., RRT*/A* (with replanning) for 'optimistic' path generation, and HJ reachability for robust planning. We illustrate how our method acts as an adaptive intermediary, providing a computationally tractable algorithm that nonetheless maintains improved safety properties relative to purely optimistic planners.

### 5.1 Simulation Experiments

**Centerline Experiment overview.** In the simulated environment, a racing vehicle (using double integrator dynamics) observes all obstacles within a fixed sensor range (Fig. 2). The nominal dynamics are perturbed by disturbances with varying levels of structure (described further below). A "centerline" environment shown in Fig. 2 is utilized to demonstrate key safety and adaptivity criteria. The nominal trajectory is fixed to be a straight path through obstacles, requiring the obstacle avoidance behavior to emerge via the online adaptation of Alg. 1. For speed, the implementation of the environment and algorithm is set up in JAX [56], using Deluca [57] as a framework for the control-theoretic simulation environment. Each simulation takes 1-2 minutes to run on a single CPU with $H = 10$, $T \approx 1000$.

**Comparison with baselines.** We utilize a HJ reachability planner [58] to generate robust trajectories, as well as a kinodynamic RRT* implementation [59] to generate "optimistic" plans (replanned at 1 Hz, with a path-following controller applied at 5 Hz). We demonstrate two key behaviors: (1) better performance than HJ methods with respect to LQR costs when disturbances are "benign;" (2) better performance than RRT* when disturbances are adversarial. For each algorithm both stochastic ['Rand'] and non-stochastic (sinusoidal ['Sin'] and adversarial ['Adv']) disturbance profiles are tested, and metrics for both the safety (number of collisions) and performance (LQR state and input costs) are collected for runs spanning 50 centerline obstacles. Results are presented for each algorithm and disturbance profile in Table 1. For space considerations, figures illustrating trajectories of simulated runs referenced below are deferred to Supp. D.

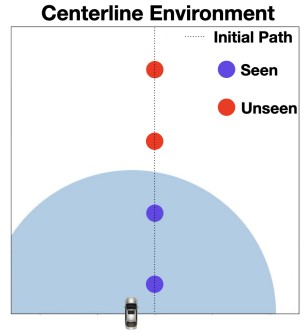

**Centerline Environment**

Figure 2: Centerline environment. The nominal path passes vertically through obstacles; the sensor range is denoted by the blue shaded region.

| Dist. / Plan | Rand | Sin | Adv |
|---|---|---|---|
| RRT* | **0.27** ± 0.05 
 0.60 | — 
 1.00 | — 
 1.00 |
| OLC (ours) | 0.51 ± 0.09 
 0.06 | **0.49** ± 0.13 
 0.04 | 1.57 ± 0.65 
 0.26 |
| HJ Plan | 0.55 ± 0.05 
 0.00 | 0.59 ± 0.14 
 0.00 | **1.01** ± 0.03 
 0.00 |

Table 1: Planner performance for each disturbance type. Costs are given in terms of linear-quadratic (LQ) costs (top) and fraction of failures (bottom). Best-performing cases for each column are **bold**. LQ costs are only computed for successful passes; as such, RRT* is intentionally blank for two entries.

Several aspects of Table 1 reflect the expected behavior of each algorithm. RRT* follows efficient paths, but fails to handle disturbances effectively, with a high collision rate. Second, HJ paths are robust, but performance improvements are limited as 'adversariality' is reduced, and the method does not adapt to disturbance structures. In particular, the sinusoid case incentivizes the racer to pass the obstacles on a specific side; the HJ planner passes on each side in equal proportion In contrast, OLC adapts to the structure of the sinusoidal disturbances to take the "easier route" with lower cost.

As shown in Table 1, OLC significantly reduces collisions across all disturbance profiles relative to RRT* planning, while also reducing control usage and state costs relative to HJ planners. This intermediate solution also provides computational speed-up to HJ, allowing it to be run more efficiently online and allow for feedback on sensory information (as opposed to the privileged map information that HJ requires in Sec. 5.2). Finally, OLC was tested on several other environments, including with dynamic obstacles – for further details, see Supp. D.

### 5.2 Hardware Experiments

**Experiment overview.** For our hardware experiments, we use the Unitree Go1 quadruped robot, shown in Fig. 1. The robot is equipped with LIDAR and an inertial measurement unit (IMU) which enable obstacle detection and localization using LIO-SAM [60]. The robot's task is to traverse a dense course of cylindrical obstacles, while encountering time delays and residual, nonlinear dynamics. The plant is modeled as a Dubins' car; the equations of motion are included in Supp. C. The high-level inputs are then translated to joint-level torque commands by the robot's low-level controller. We note that the noisy estimates on hardware of both the state and the relative obstacle positions will further demonstrate the OLC algorithm's robustness to noise in the inputs to Alg. 1, including in the reconstructed disturbances and in the estimated state.

**Controller architecture.** All sensing and computation is performed onboard the robot. We use a Euclidean clustering algorithm for obstacle detection [61] based on the LIDAR measurements. This detection algorithm runs onboard the robot and provides updated obstacle locations to the controller in real time (not all obstacles are initially sensed due to occlusions and larger distances). At each replanning step, the robot generates a nominal set of waypoints from the estimated state and detected obstacles using an A* algorithm [62] over a discretization of the traversable space, which is converted to a continuous path by a down-sampled smoothing spline. We then wrap the additional OLC feedback controller ($H = 5$, $T = 40$) of the form presented in Eqn. 3 using Alg. 1. The use of A* in lieu of RRT* was undertaken to ensure smoother resulting paths for the quadruped. We justify the change by noting that due to discretization A* should only be *equally or more robust* than RRT* (using equivalent margin), thereby *reducing* the apparent benefits of the OLC module.

**Baselines.** We compare OLC with the baselines used for the simulated experiments (Sec. 5.1), substituting A* for RRT* to improve speed and stability of the nominal planner. Concretely, the first baseline uses A* in a receding horizon re-planning scheme with the nominal dynamics model

and obstacle padding of 0.25m to account for the Go1's size. The robot executes the control inputs provided by the planner, replanning at 1 Hz and taking actions at 4 Hz. The second baseline is a robust controller that is generated using Hamilton-Jacobi (HJ) reachability. This baseline is provided an *a priori* map of the obstacles, and uses a provided model of the disturbances, taking into account velocity-dependent effects. This second baseline acts as a safety 'oracle,' with optimal trajectories synthesized offline from the true map. As before, we demonstrate that our algorithm improves the safety of a pure A* (with margin) while providing shorter and more intuitive paths than HJ methods.

| Metric
Planner | Path Length (m) | Max Deviation (m) | Collisions |
|:---:|:---:|:---:|:---:|
| A$^*$ | $\mathbf{10.61} \pm 0.40$ | $\mathbf{0.62} \pm 0.33$ | 12 |
| Online (OLC) | $\mathbf{10.59} \pm 0.41$ | $\mathbf{0.63} \pm 0.28$ | 7 |
| HJ Plan | $11.13 \pm 0.45$ | $1.16 \pm 0.57$ | **0** [simulated] |

Table 2: Planner performance along several metrics for the hardware examples. We observe that our method ('Online') performs very similarly to A* in terms of path efficiency while reducing collisions by over 40%. Note that HJ methods choose significantly longer paths to account for worst-case uncertainty; HJ was not run on hardware and the optimal paths were calculated offline using the true obstacle positions.

**Results.** We run physical experiments using A$^*$ and OLC for 21 runs each, consisting of three trials over seven obstacle layouts (see the supplemental video and Supp. C for additional details). In each instance, the robot is required to traverse 10m forwards. For each layout, the optimal HJ path is analyzed offline; for all runs, a 'crash' is defined as any contact made with an obstacle. As shown in Table 2, HJ methods choose an overly-conservative route that is generally safe but inefficient. Conversely, A$^*$ – even with obstacle padding of 0.25m – does not sufficiently account for the disturbances, yielding a significant rate of crashes. However, in contrast to HJ, A$^*$ and OLC take significantly shorter paths (note that the shortest possible path is at minimum 10m in length). However, our algorithm reduces the number of collisions by nearly half versus the naive obstacle-padded A$^*$ approach, a result significant at $p = 0.1$ using a Boschloo exact test (see Supp. C). Importantly, while OLC was run onboard the robot in real time, the HJ methods required several seconds for each computation of the backward reachable set, and could not be run online.

## 6  Conclusion and Limitations

We develop a regret minimization framework for the problem of online obstacle avoidance. In contrast to prior approaches that either assume worst-case realization of uncertainty or a given stochastic model, we utilize techniques from online learning in order to *adaptively* react to disturbances and obstacles. To this end, we prove regret bounds that demonstrate that our obstacle avoidance policy is comparable to the best policy *in hindsight* from a given class of closed-loop policies. Simulation and hardware experiments demonstrate that our approach compares favorably with baselines in terms of computational efficiency and performance with varying disturbances and obstacle behaviors.

Some limitations that we hope to address in future work include the limited classes of applicable dynamics (though our hardware experiments apply the method to time-varying linear dynamics) and obstacle geometries. Additionally, the cost function does not have multi-time-step lookahead (i.e., MPC-like). Additional lookahead would yield better foresight in the planning stage and likely improve performance. Finally, because the algorithm relies on solutions to optimization problems that run real-time, ensuring stability of the optimization procedure and automating selection of hyperparameters (those used are satisficing but likely suboptimal) would be useful extensions that may further improve the runtime performance.

**Acknowledgments**

This work was partially funded by the NSF Career Award No. 2044149. Additionally, this material is based upon work supported by the National Science Foundation Graduate Research Fellowship under Grant No. DGE-2039656. Any opinion, findings, and conclusions or recommendations expressed in this material are those of the authors(s) and do not necessarily reflect the views of the National Science Foundation.

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

# A  Full Regret Proof

## A.0  Outline of Proof

1. Reduce path planner to linear dynamical systems model
2. Demonstrate that a general instance of Alg. 2 is a trust region problem
3. Reduce Eqn. 9 in Alg. 1 to an instance of Alg. 2
4. Justify necessary analogous quantities in our problem to those of the proofs in [40, 38]
5. Apply the results for Nonconvex FPL with Memory from [40] to Alg. 1 to obtain the regret bound

## A.1  Reduction of Path Planned case to Standard Controls case

Assume that the planner devises a nominal path (denoted with a $(\bar{\cdot})^0$ notation) in coordinates $\mathbf{x}$ and inputs $\mathbf{u}$: so the path $\mathcal{P}$ is fully specified as $\mathcal{P} = \{\bar{\mathbf{x}}_t^0, \bar{\mathbf{u}}_t^0\}_{t=0}^T$. Assume that the path is chosen so that at every $\mathbf{x}$ on or near the path, the following dynamics hold around perturbations of the path:

$$\mathbf{x}_t - \bar{\mathbf{x}}_t^0 = A(\mathbf{x}_{t-1} - \bar{\mathbf{x}}_{t-1}^0) + B(\mathbf{u}_{t-1} - \bar{\mathbf{u}}_{t-1}^0) + D\mathbf{w}_{t-1}. \tag{10}$$

Using this change of coordinates, we can essentially negate the path and study the relevant perturbation dynamics $\delta\mathbf{x}_t := \mathbf{x}_t - \bar{\mathbf{x}}_t^0$ and $\delta\mathbf{u}_t := \mathbf{u}_t - \bar{\mathbf{u}}_t^0$, we recover the desired equation:

$$\delta\mathbf{x}_t = A\delta\mathbf{x}_{t-1} + B\delta\mathbf{u}_{t-1} + D\mathbf{w}_{t-1}. \tag{11}$$

For shorthand, we will define $\mathbf{x} := \delta\mathbf{x}$ and $\mathbf{u} = \delta\mathbf{u}$ to ease exposition, remembering that they represent perturbations from the nominal path. Intuitively, this is a reasonable model for 'quasi-static' systems (e.g., a drone or car or aircraft using path planning for non-aggressive maneuvers).

## A.2  Algorithm 2 is a Trust Region Solver

The proof that Alg. 2 is a trust region solver is given in Supp. B.1- B.3.

## A.3  Algorithm 2 Solves Equation 9

The proof that Eqn. 9 is a special instance of admissible arguments to Alg. 2 is shown in Supp. B.4.

## A.4  Technical Notes

### A.4.1  Continuity and Conditioning Parameters

We begin with an analysis of the Lipschitz constant for the approximate cost functions (this will follow a similar path to [40]).

First, note that the diameter of the decision set is $2D_M$ and that the gradient of the quadratic cost above is $\nabla_{\mathbf{m}}\ell_t = (P + P^T)\mathbf{m} + \mathbf{p}$. As such,

$$\begin{aligned}
L &:= \max_{\mathbf{m},t} \{\|\nabla_{\mathbf{m}}\ell_t(\mathbf{m})\|_\infty\} \\
&\leq \max_{\mathbf{m},t} \{(\|P\|_1 + \|P\|_\infty)D_M + \|\mathbf{p}\|_\infty\} \\
&\leq 2Hd_w RD + R
\end{aligned}$$

We consider as well a bound on the conditioning number of the optimization problem. Because the size of the optimization grows linearly in time, the condition number grows at most linearly as well. Therefore, the run-time of the algorithm is polynomial (neither the condition number nor the dimension grows too rapidly).

Finally, we note bounds on the elements of $P$ and $\mathbf{p}$ in the trust region instance. The bounds on costs, states, inputs, and disturbances together imply that the elements of $P_t$ are bounded by $C_u^2\kappa^2\xi$, and the elements of $\mathbf{p}$ are bounded by $C_u^2\kappa^2\xi\beta$ (this again follows [40]).

### A.4.2 Truncated State Approximation

The idea of this proof follows directly from [40]; however, we show the proof in detail because in our case the truncated state history affects the resulting vectors $\mathbf{a}_j$ in the optimization, leading to a different instantiation of the problem.

To give a sense of the added subtlety for obstacle avoidance, observe that in certain scenarios, small perturbations in the observed relative obstacle positions could yield large changes in the optimal policy. For example, imagine that there is one obstacle, located directly on the centerline of the nominal planned motion. Then a small perturbation of the obstacle to the right makes the optimal action "Left," while a small perturbation of the obstacle to the left makes the optimal action "Right."

This phenomenon is not a problem in the regret outline because, while the optimal decision is fragile, the loss incurred of choosing incorrectly is bounded by the quadratic (and therefore, continuous) nature of the cost functions themselves.

For the dynamics and control we have assumed that

$$
\begin{aligned}
x_{t+1} &= A\mathbf{x}_t + B\mathbf{u}_t + D\mathbf{w}_t \\
\mathbf{u}_t &= K\mathbf{x}_t + \mathbf{b}_t + M_t\tilde{\mathbf{w}}_t \\
&= K\mathbf{x}_t + \sum_{i=1}^{H} M_t^{[i]}\mathbf{w}_{t-i},
\end{aligned}
\tag{12}
$$

where the bias is included by one-padding the disturbance vector. For simplicity we will omit the explicit bias from ensuing analysis; in all cases it can be understood to be incorporated into the measured disturbance. We can then show (as in [40]) that the state can be expressed as the sum of disturbance-to-state transfer function matrices $\Psi_{t,i}$:

$$
\mathbf{x}_{t+1}^{\mathcal{A}} = \tilde{A}^{H+1}\mathbf{x}_{t-H}^{\mathcal{A}} + \sum_{i=0}^{2H} \Psi_{t,i}\mathbf{w}_{t-i}, \text{ where}
$$

$$
\tilde{A} = A + BK \text{ and}
$$

$$
\Psi_{t,i} = \tilde{A}^i D \mathbb{1}[i \le H] + \sum_{j=0}^{H} \tilde{A}^j B M_{t-j}^{[i-j]} \mathbb{1}[i-j \in \{1,...,H\}].
$$

We define the state estimate and cost as

$$
\mathbf{y}_{t+1} := \sum_{i=0}^{2H} \Psi_{t,i}\mathbf{w}_{t-i}
$$

$$
\ell_t(M_{t-H:t}) = c_t(\mathbf{y}_{t+1}(M_{t-H:t}), \tilde{\mathbf{u}}_t)
$$

where $\tilde{\mathbf{u}}_t = M_t\tilde{\mathbf{w}}_t$ (the residual input on top of the closed-loop controller).

Now, assume that $\|\tilde{A}\| \le 1 - \gamma$, that $\|\tilde{A}\|, \|B\|, \|D\|, \|K\| \le \beta$, and that for all $t$ it holds that $\|w_t\| \le C_w$, $\|u_t\| \le C_u$, and $\|Q_t\|, \|R_t\| \le \xi$. Then we can show that the approximation error of the costs is sufficiently small. Let the condition number be defined as $k = \|\tilde{A}\|\|\tilde{A}^{-1}\|$.

### A.4.3 Bounding the States Along a Trajectory

Note that $\tilde{\mathbf{u}}_t = M_t\tilde{\mathbf{w}}_t$; this implies that $\|\tilde{\mathbf{u}}_t\| \le HDC_w$. This implies further that $\|B\tilde{\mathbf{u}}_t + D\mathbf{w}_t\| \le 2\beta HDC_w$ by the triangle inequality. Assuming that there exists $\tau$ such that

$$
\|\mathbf{x}_\tau\|_2 \le \frac{2\beta HDC_w}{\gamma},
$$

we have that for every $t > H + \tau + 1$, $\|\mathbf{x}_{t-H-1}^{\mathcal{A}}\|_2 \le \frac{2\beta HDC_w}{\gamma}$. (WLOG, we can assume the initial state $\mathbf{x}_0$ is bounded in this domain - that is, that the assumption is satisfied with $\tau = 0$; the region defined above is the long-term reachable set of the state $\mathbf{x}_t$ driven by bounded disturbances $\mathbf{w}_t$ and (implicitly bounded) residual inputs $\tilde{\mathbf{u}}_t$ [the norm is limited by the stability parameter $\gamma$ of the closed-loop $\tilde{A}$-matrix]).

### A.4.4 Bounding the Change in Costs

Now, we analyze the change in costs

$$|c_t(\mathbf{x}_{t+1}^{\mathcal{A}}, \tilde{\mathbf{u}}_t) - \ell_t(M_{t-H:t})| = |\min_{j \in [p]} \|\mathbf{a}_{j,t} + BM_t \tilde{\mathbf{w}}_t\|_2^2 - \min_{j \in [p]} \|\hat{\mathbf{a}}_{j,t} + BM_t \tilde{\mathbf{w}}_t\|_2^2|$$

Noting the definition of $\hat{\mathbf{a}}_{j,t}$ and of $\mathbf{a}_{j,t}$, we can bound the difference between them as a function of the error in approximation of $\mathbf{x}_t$ (see [40]):

$$\mathbf{a}_{j,t} := \mathbf{p}_{j,t} - \mathbf{x}_t$$
$$\implies \hat{\mathbf{a}}_{j,t} - \mathbf{a}_{j,t} = (\mathbf{p}_{j,t} - \hat{\mathbf{x}}_t) - (\mathbf{p}_{j,t} - \mathbf{x}_t)$$
$$= \mathbf{x}_t - \hat{\mathbf{x}}_t$$
$$\implies \|\hat{\mathbf{a}}_{j,t} - \mathbf{a}_{j,t}\|_2 = \|\mathbf{x}_t - \hat{\mathbf{x}}_t\|_2$$
$$\leq k C_x e^{-\gamma H}$$

Now, we argue that the loss incurred due to the noise in $\hat{\mathbf{x}}_t$ is less than simply twice the change in cost due to the error in $\hat{\mathbf{a}}_{j,t}$. Let $\hat{j}^* := \arg\min_{j \in [p]} \{\|\hat{\mathbf{a}}_{j,t} - BM_t \tilde{\mathbf{w}}_t\|_2^2\}$. Let $j^*$ be defined analogously. If $j^* = \hat{j}^*$, then the difference in cost is less than or equal to the extra loss incurred by the error in $\hat{\mathbf{a}}$. If $j^* \neq \hat{j}^*$, then it is possible that the true 'binding obstacle' was biased away, and that the 'guessed' binding obstacle was 'biased towards'; therefore, the cost error is possibly due to deviations up to twice the error in the $\hat{\mathbf{a}}_{j,t}$ vectors. This means that, defining $\delta_t$ such that $\|\delta_t\|_2 = 2\|\mathbf{x}_t - \hat{\mathbf{x}}_t\|_2$, we have that the following holds:

$$\Delta = |c_t(\mathbf{x}_{t+1}^{\mathcal{A}}, \tilde{\mathbf{u}}_t) - \ell_t(M_{t-H:t})| = |\min_{j \in [p]} \|\mathbf{a}_{j,t} + BM_t \tilde{\mathbf{w}}_t\|_2^2 - \min_{j \in [p]} \|\hat{\mathbf{a}}_{j,t} + BM_t \tilde{\mathbf{w}}_t\|_2^2|$$

$$\leq \max_{\delta_t : \|\delta_t\|_2 \leq 2kC_x e^{-\gamma H}} \left\{ \|(\hat{\mathbf{a}}_{j,t} + \delta_t) + BM_t \tilde{\mathbf{w}}_t\|_2^2 - \|\hat{\mathbf{a}}_{j,t} + BM_t \tilde{\mathbf{w}}_t\|_2^2 \right\}$$

$$= \delta_t^T \delta_t + 2\delta_t^T \hat{\mathbf{a}}_{j,t} - 2\delta_t^T (BM_t \tilde{\mathbf{w}}_t)$$

$$\leq \|\delta_t\|_2^2 + 2(C_x + \|\delta_t\|_2)\|\delta_t\|_2 + 2\|\delta_t\|_2 C_w \beta D_M$$

$$= 3\|\delta_t\|_2^2 + 2C_x \|\delta_t\|_2 + 2C_w \beta D_M \|\delta_t\|_2$$

$$\leq 5C_x \|\delta_t\|_2 + 2C_w \beta D_M \|\delta_t\|_2$$

$$\leq 5(k^2 C_x^2 e^{-\gamma H}(1 + \beta D_M C_w)).$$

Letting $H = \lceil \gamma^{-1} \log \left(5k^2 C_x(1 + \beta D_M C_w)T\right) \rceil$, we have that

$$\Delta \leq \frac{C_x}{T}.$$

**Remark 3.** *Recursive Definition of $H$ and $C_x$:*

*Currently, there is a recursive nature to the definition of $H$ and $C_x$; $H :=$ $\lceil \gamma^{-1} \log\left(5k^2 C_x (1 + \beta D C_w) T\right) \rceil$ and $C_x := \frac{2\beta H D C_w}{\gamma}$. However, this is not problematic because the definitions will have a solution (that can be found efficiently); namely:*

$$H \geq c_1 \log\left(c_2 C_x\right)$$
$$C_x = k_1 H$$

$$\implies H \geq c_1 \log\left(c_2 k_1 H\right)$$

*And for any $c_1, c_2, k_1 \in \mathbb{R}^+$ and fixed $T > 0$, there exists a positive integer $H$ such that the above result holds (e.g., following from the fact that $\log H = o(H)$). Further, the resulting $H$ will not be too large wrt $T$ for sufficiently large $T$ (e.g., large enough $T$ to overcome the constants).*

### A.5   Finalizing the Regret Bound

#### A.5.1   Apply Nonconvex Memory Follow-the-Perturbed-Leader

This result is from [40], Theorem 13 (Corollary 14 gives an equivalent result to our setting in the asymptotic regret behavior; our optimal choice of $\eta$ and $\epsilon$ differs slightly).

#### A.5.2   Completing the Bound

Finally, we use Alg. 1 (which acts as an efficient $\epsilon$-oracle) with an approximate trust region implementation of our desired optimization problem (Alg. 2) acting as a subroutine, in order to compose the regret components into a complete bound.

$$
\begin{aligned}
\text{Regret}(\mathcal{A}) &:= \max_{M \in \Pi} \sum_{t=H}^{T} c_t(x_t^M, \tilde{u}_t(M)) - \sum_{t=H}^{T} c_t(x_t^{\mathcal{A}}, \tilde{u}_t(\mathcal{A})) \\
&\leq \max_{M \in \Pi} \sum_{t=H}^{T} \left(f_t(M, M, ..., M) + \frac{C_x}{T}\right) - \sum_{t=H}^{T}\left(f_t(M_{t-H:t}) + \frac{C_x}{T}\right) \\
&= \left[\max_{M \in \Pi} \sum_{t=H}^{T} f_t(M, M, ..., M) - \sum_{t=H}^{T} f_t(M_{t-H:t})\right] + \mathcal{O}(\log T) \\
&\leq \tilde{\mathcal{O}}(\text{poly}(\mathcal{L})\sqrt{T})
\end{aligned}
$$

(13)

To clarify the steps: the second line incorporates the approximation error from Section A.4.2 (which is logarithmic in $T$, as noted in the third line) and the final line follows from the Nonconvex Memory FPL result of [40].

# B  Convex-Concave Game: Algorithm and Correctness

For completeness and easier reference, we include a copy of Alg. 1 below. To improve clarity, references to equations within this section (Supp. B) will use their numbering as given in Supp. B, rather than in the main text. The key technical results of this section are to demonstrate: (1) that Alg. 2 is a trust region instance that can be solved efficiently, and (2) that the optimization procedure in Eqn. 15 is solved correctly and efficiently by the trust region procedure Alg. 2. Given these results, our obstacle avoidance algorithm will be computationally efficient and attain low regret.

---

**Algorithm 1** Online Learning Control (OLC) for Obstacle Avoidance
***

**Input**: Observed obstacle positions $\{\mathbf{p}_0^j\}_{j=1}^K$, history length $H = \mathcal{O}(\log T)$.
**Input**: Full horizon $T$, algorithm parameters $\{\eta, \epsilon, \lambda\}$, initial state $\mathbf{x}_0$.
**Input**: Open-loop plan: $\bar{\mathbf{u}}_t^o$ for $t = 1, ..., T$.
**Initialize**: Closed-loop correction $M_0^{[1:H]}$, fixed perturbation $P_0 \sim \mathrm{Exp}(\eta)^{d_u \times H d_w}$ .
**Initialize**: Play randomly for $t = \{0, ..., H-1\}$, observe rewards, states, noises, and obstacles.
**for** $t = H...T-1$ **do**
  Play $M_t^{[1:H]}$, and observe state $\mathbf{x}_{t+1}$ and obstacles $\{\mathbf{p}_{t+1}^j\}_{j \in [k]_{t+1}}$.
  Reconstruct disturbance $\mathbf{w}_t$ using observed $\mathbf{x}_{t+1}$.
  Construct the reward function in $\tilde{M}$:

$$\ell_{t+1}(\tilde{M}) = \min_{j \in [k]} \left\{ \|\mathbf{x}_{t+1}^{\tilde{M}} - \mathbf{p}_{t+1}^j\|_2^2 \right\} - \|\mathbf{x}_{t+1}^{\tilde{M}}\|_Q^2 - \|\tilde{\mathbf{u}}_t^{\tilde{M}}\|_R^2. \tag{14}$$

  Receive realized reward $l_{t+1}(M_t^{[1:H]})$.
  Solve for 'perturbed leading policy' $M_{t+1}^{[1:H]}$ as the solution to:

$$\arg\max_{\|\tilde{M}\| \leq D_M} \left\{ \sum_{\tau=1}^{t+1} \ell_\tau(\tilde{M}) + \lambda(\tilde{M} \bullet P_0) \right\}. \tag{15}$$

**end for**

---

**Algorithm 2** (General) Hidden-Convex Formulation for Objective in Eqn. 9
***

**Input**: Set of vectors $\{\mathbf{a}_j^{[\tau]}\}_{j=1,\tau=1}^{k,H_t}$, matrix $B$, vectors $\mathbf{b}, \mathbf{b}_0$, time history $H_t \leq t$
**Input**: Iterations $N$, learning rate $\eta$, approx. error $\epsilon$, perturbation $P_0$, diameter $D_M$.
**Initialize**: Vector $\mathbf{c}_0^{[\tau]} = \frac{1}{k}\mathbb{1}^k, \tau = 1, \ldots, H_t$.
**for** n=0...N **do**
  (1) Solve for $M_n$

$$M_n = \arg\max_{\|M\| \leq D_M} \left\{ \sum_{\tau=1}^{H_t} \sum_{j=1}^k \mathbf{c}_n(j)^{[\tau]} \|\mathbf{a}_j^{[\tau]} + BM\mathbf{b}^{[\tau]}\|_2^2 \right.$$
$$\left. - \|\mathbf{b}_0^{[\tau]} + BM\mathbf{b}^{[\tau]}\|_Q^2 - \|M\mathbf{b}^{[\tau]}\|_R^2 + \lambda(M \bullet P_0) \right\}. \tag{16}$$

  (2) Update $\mathbf{c}_{n+1}$

$$\mathbf{c}_{n+1}^{[\tau]} = \prod_{\Delta_k} \left[ \mathbf{c}_n^{[\tau]} e^{-\eta \nabla_{\mathbf{c}} \left( \sum_j \mathbf{c}_n^{[\tau]}(j) \|\mathbf{a}_j^{[\tau]} + BM\mathbf{b}^{[\tau]}\|_2^2 \right)} \right], \forall \tau \in \{1, \ldots, H_t\}. \tag{17}$$

**end for**
**return** $M_N$

---

## B.1 Non-convex Memory FPL for Obstacle Avoidance

Intuitively, Alg. 1 operates by updating the gain matrices $M_{t+1}^{[1:H]}$ via counterfactual reasoning: *in hindsight*, given the actual observed disturbances and obstacle locations, what gain matrices *would have resulted* in good performance (in terms of obstacle avoidance and the state-input penalties)? In Supp. A, we demonstrate that this algorithm results in low regret as formalized in Eqn. 5 of the main text. For reference: Eqn. 9 (main text) corresponds to Eqn. 15 (Supp. B) henceforth; similarly, Eqn. 8 (main text) corresponds to Eqn. 14 (Supp. B).

## B.2 Efficient Solution of Eqn. 15 (Part 1): Reduction of Alg. 2 to Trust Region Instance

We now prove that Alg. 2 does indeed solve Eqn. 9. Consider the relaxed optimization problem

$$\max_{M \in \mathcal{M}} \sum_{\tau=1}^{H_t} \sum_j \lambda_j^{[\tau]} \|\mathbf{a}_j^{[\tau]} + BM\mathbf{b}^{[\tau]}\|_2^2 \tag{18}$$

We will first describe some useful quantities and (physically-motivated) assumptions. The physical quantities of interest have the following characteristics: $\mathbf{x} \in \mathbb{R}^{d_x}$, $\mathbf{u} \in \mathbb{R}^{d_u}$, and $\mathbf{w} \in \mathbb{R}^{d_w}$.

**Assumption 4.** $B \in \mathbb{R}^{d_x \times d_u}$, with $d_u \leq d_x$, and rank $(B) = d_u$. *This corresponds to the following physical assumptions: (1) there are no more inputs than states, and (2) there are no 'extraneous' inputs (if there are such inputs, then we can find a minimal realization of $B$ and wlog set extraneous inputs to always be zero). Similarly, if (1) fails, then we can remove extraneous inputs by again setting them equal to zero uniformly (just as in (2)).*

The dimension of several other variables of interest are: $\mathbf{b} \in \mathbb{R}^{H d_w}$ and for the decision variable $M$, $M \in \mathbb{R}^{d_u \times H d_w}$. We want to show that the optimization in Eqn. 18 is equivalent to a convex trust region problem, which is efficiently solvable.

Further, let the quantity $B^T B$ have a singular value decomposition denoted by:

$$B^T B = U^T \Lambda U.$$

We see that this decomposition has an orthogonal $U \in \mathbb{R}^{d_u \times d_u}$ and positive definite $\Lambda$ because rank$(B) = d_u$ and thus the symmetric $B^T B \in \mathbb{R}^{d_u \times d_u}$ has $B^T B \succ 0$.

Now, consider the problem of Eqn. 18, assuming that $\forall \tau \in \{1, \ldots, H_t\}$, there are fixed $\lambda^{[\tau]} \in \Delta_p, B, \{\mathbf{a}_j^{[\tau]}\}_{j=1}^k, \mathbf{b}^{[\tau]}$. For now, choose a single time element, notated as $[i]$. For simplicity, we will only include this notation at the beginning and end, where we sum the result back together. We rearrange the objective for this case as follows:

$$\begin{aligned}
\text{OBJ}_{\text{partial}} &= \sum_j \lambda_j^{[i]} \|\mathbf{a}_j^{[i]} + BM\mathbf{b}^{[i]}\|_2^2 \\
&= \sum_j \lambda_j (\mathbf{a}_j + BM\mathbf{b})^T (\mathbf{a}_j + BM\mathbf{b}) \\
&= \sum_j \lambda_j (\mathbf{a}_j^T \mathbf{a}_j + 2\mathbf{a}_j^T BM\mathbf{b} + \mathbf{b}^T M^T B^T BM\mathbf{b}) \\
&\equiv \sum_j \lambda_j (2\mathbf{c}_j^T M\mathbf{b} + \mathbf{b}^T M^T U^T \Lambda UM\mathbf{b})
\end{aligned}$$

Here, we are searching for the argmax $M^*$, so the $\mathbf{a}_j^T \mathbf{a}_j$ is irrelevant. Further, we have defined $\mathbf{c}_j^T = \mathbf{a}_j^T B$. Now, let $M_r := UM$, and decompose $M_r = [\mathbf{m}_1^T; \mathbf{m}_2^T; ...; \mathbf{m}_{d_u}^T]$. The optimization

objective is now:

$$\text{OBJ}_{\text{partial}} = \sum_j \lambda_j \|\mathbf{a}_j + BM\mathbf{b}\|_2^2$$

$$\equiv \sum_j \lambda_j (2\mathbf{c}_j^T M\mathbf{b} + \mathbf{b}^T M^T U^T \Lambda U M\mathbf{b})$$

$$= \sum_j \lambda_j (2\mathbf{c}_j^T M\mathbf{b} + \mathbf{b}^T M_r^T \Lambda M_r \mathbf{b})$$

$$= [\sum_j \lambda_j (2\mathbf{c}_j^T M\mathbf{b})] + \mathbf{b}^T M_r^T \Lambda M_r \mathbf{b}$$

The last simplification follows from the fact that $\sum_j (\lambda_j) = 1$ (because $\lambda \in \Delta_{d_u}$). Now, using our knowledge of the diagonal nature of $\Lambda$ and the column partition of $M_r$, we can see that

$$M_r^T \Lambda M_r = \sum_{j=1}^{d_u} \sigma_j^2 \mathbf{m}_j \mathbf{m}_j^T$$

Substituting into the OPT formulation, we can further simplify all the way to the desired form:

$$\text{OBJ}_{\text{partial}} = \sum_i \lambda_i \|\mathbf{a}_i + BM\mathbf{b}\|_2^2$$

$$\equiv \Big[ \sum_i \lambda_i (2\mathbf{c}_i^T M\mathbf{b}) \Big] + \mathbf{b}^T M_r^T \Lambda M_r \mathbf{b}$$

$$= \Big[ \sum_i \lambda_i (2\mathbf{c}_i^T M\mathbf{b}) \Big] + \mathbf{b}^T (\sum_{j=1}^{d_u} \sigma_j^2 \mathbf{m}_j \mathbf{m}_j^T) \mathbf{b}$$

$$= \Big[ \sum_i \lambda_i (2\mathbf{c}_i^T U^T M_r \mathbf{b}) \Big] + (\sum_{j=1}^{d_u} \sigma_j^2 \mathbf{b}^T \mathbf{m}_j \mathbf{m}_j^T \mathbf{b})$$

$$= 2\Big[ \sum_i \lambda_i (\sum_j \tilde{c}_{i,j} \mathbf{m}_j^T \mathbf{b}) \Big] + (\sum_{j=1}^{d_u} \sigma_j^2 \mathbf{m}_j^T \mathbf{b}\mathbf{b}^T \mathbf{m}_j)$$

$$= \sum_j \Big[ (2\sum_i \lambda_i \tilde{c}_{i,j}) \mathbf{b}^T \Big] \mathbf{m}_j) + (\sum_{j=1}^{d_u} \mathbf{m}_j^T (\sigma_j^2 \mathbf{b}\mathbf{b}^T) \mathbf{m}_j)$$

$$= \mathbf{m}^T P\mathbf{m} + \mathbf{p}^T \mathbf{m}.$$

where $\mathbf{m}$ is a vector concatenation of the transposed rows of $M_r$. Now, to combine the results over time, utilizing the convexity-preserving property of function addition, we simply reintroduce the $[i]$-indexing and sum:

$$\text{OBJ}_{\text{partial}} = \sum_j \lambda_j^{[i]} \|\mathbf{a}_j^{[i]} + BM\mathbf{b}^{[i]}\|_2^2$$

$$= \mathbf{m}^T P^{[i]} \mathbf{m} + (\mathbf{p}^{[i]})^T \mathbf{m}$$

$$\implies \text{OBJ}_{\text{full}} = \sum_i \sum_j \lambda_j^{[i]} \|\mathbf{a}_j^{[i]} + BM\mathbf{b}^{[i]}\|_2^2$$

$$= \sum_i \mathbf{m}^T P^{[i]} \mathbf{m} + (\mathbf{p}^{[i]})^T \mathbf{m}$$

$$= \sum_i \mathbf{m}^T P\mathbf{m} + \mathbf{p}^T \mathbf{m}$$

Once we solve for $\mathbf{m}^*$ using a trust region solver, we unpack it into $M_r^*$, and get $M^* = U^T M_r^* (= U^T (UM^*) = M^*)$ as desired. Further, we can translate a norm bound on $M$ into an equivalent one on $\mathbf{m}$ (at least, for appropriate choice of norm bound - e.g., the Frobenius norm on $M$ becomes the 2-norm on $\mathbf{m}$).

## B.3 Solving the FPL Sub-Problem

In order to feasibly engage the obstacle avoidance algorithm, it is necessary to solve for optimal solutions to the objective in Eqn. 15, which is an instance of the following max-min problem:

$$\underset{\|M\| \leq D_M}{\arg\max} \sum_{\tau=1}^{t+1} \min_j \|\mathbf{a}_j^{[\tau]} + BM\mathbf{b}^{[\tau]}\|_2^2 - \|\mathbf{b}_0^{[\tau]} + BM\mathbf{b}^{[\tau]}\|_Q^2 - \dots$$

$$\dots - \|M\mathbf{b}^{[\tau]}\|_R^2 + \lambda(M \bullet P_0),$$

where $\mathbf{a}_j^{[\tau]}, B, \mathbf{b}_0^{[\tau]}, \mathbf{b}^{[\tau]}$ depend on the dynamics (Eqn. 4, main text) and the obstacle locations. The resulting algorithm is shown in Alg. 2, which converges to $M_{t+1}^{[1:H]}$ in Eqn. 15 of Alg. 1.

Finally, we need to demonstrate that Alg. 2 will converge to a pair $\{\mathbf{c}_N, M_N\}$ that corresponds to the optimal solution of Eqn. 15. This follows immediately from Theorem 7, Part II of [54]. We have an instance of a repeated game in which an optimization oracle efficiently solves Eqn. 16, and then the low-regret exponentiated gradient algorithm [63] iteratively updates $\mathbf{c}_n$ (Eqn. 17). Again, we utilize the fact that the operation of function addition preserves, respectively, the convexity and concavity properties of pairs of operand functions as a necessary tool to allow for the results to still hold when applying the summation over time steps.

## B.4 Efficient Solution of Eqn. 15 (Part 2): Reduction of Obstacle Avoidance to Alg. 2

This proof works as a reduction, where we show that the obstacle avoidance problem (Eqn. 15) constitutes a particular set of inputs to the general formulation of Alg. 2. Recall that at time $t$, the algorithm $A$ has access to the state trajectory $\{\mathbf{x}_\tau^A\}_{\tau=1}^t$, the disturbance history $\{\mathbf{w}_\tau\}_{\tau=1}^{t-1}$, and the sets of sensed obstacles $\{\mathbf{p}_\tau^j\}_{\tau=1}^t$. For any $\tau \in \{H, ..., t\}$, the loss function can be written as an instance of Alg. 2. Specifically, let $\mathbf{a}_\tau^j = \tilde{A}\mathbf{x}_{\tau-1} + D\mathbf{w}_{\tau-1} - \mathbf{p}_\tau^j$, $\mathbf{b}_\tau = \mathbf{w}_{\tau-H:\tau-1}$, $\mathbf{b}_{0,\tau} = \tilde{A}\mathbf{x}_{\tau-1} + D\mathbf{w}_{\tau-1}$. Then, for an appropriate $\mathbf{c}_\tau \in \Delta_{k_\tau}$, the optimization problems are equivalent. Concatenating over the $\tau$-formulations, we have an instance of Eqn. 16. Specifically, for some choice of $\mathbf{c}$, we will have an equivalent problem as Eqn. 15; here $\mathbf{c}$ is an encoding – unknown *a priori* – of the relevant (nearest) obstacles at each time step.

## C Hardware Experiment Details

### C.1 Equations of Motion

The equations of motion for the high-level Go1 control are:

$$\begin{bmatrix} x_{t+1} \\ y_{t+1} \\ \psi_{t+1} \end{bmatrix} = \begin{bmatrix} x_t \\ y_t \\ \psi_t \end{bmatrix} + dt \begin{bmatrix} \cos\psi & 0 \\ \sin\psi & 0 \\ 0 & 1 \end{bmatrix} \begin{bmatrix} u_x \\ u_\psi \end{bmatrix} + \begin{bmatrix} w_{x,t} \\ w_{y,t} \\ w_{\psi,t} \end{bmatrix}, \tag{19}$$

where $u_x$ is the commanded forward velocity and $u_\psi$ is the commanded yaw rate. The disturbances $w_x, w_y$ and $w_\psi$ capture unmodeled disturbances including imperfect velocity tracking by the robot's low-level controller and deviations due to localization noise.

### C.2 Boschloo Test for Significance

In order to evaluate the improvement of OLC vs A$^*$ in our hardware experiments, we perform a Boschloo Exact Test on the outcomes of the experiment. This test (for our purposes) essentially provides a statistical evaluation of the difference in means of two Bernoulli variables in the small-data regime. In our experiments, we have a 2x2 test matrix in which OLC and A$^*$ are each run 21 times, with the number of failures a random variable depending on each algorithm's behavior subject to the realized obstacle layouts. While we do not claim that our distribution of layouts exactly matches the 'true distribution in the world,' we believe it to be similar enough such that the statistic should be meaningful here.

To be precise, the statistic encapsulates the probability, in the null hypothesis that the two means are equal (or that one mean is greater), of achieving a small sample of realizations that is *more extreme* than that observed in the true data. By choosing the alternative hypothesis "collision rate of A$^*$ is higher," a significant result (say, at $\alpha = 10\%$ or $\alpha = 5\%$ significance) would mean that we reject the null hypothesis. This, then, would provide evidence that the use of OLC meaningfully reduced the collision rate.

Using the existing scipy implementation (scipy.stats.boschloo_exact), we find that for the data presented in Table 2, the test statistic is $0.1073$ with $p = 0.074$, indicating significance at $\alpha = 0.1$ but not $\alpha = 0.05$. Though not conclusive, this provides reasonable evidence that the reported improvement in the collision rate is not due to random chance in the obstacle layout realizations.

### C.3 Obstacle Layouts

We provide a bird's eye view of the layout configurations used during the experiments in Figure 3. Additionally see the supplementary video for the physical instantiation of the layouts used in the experiments.

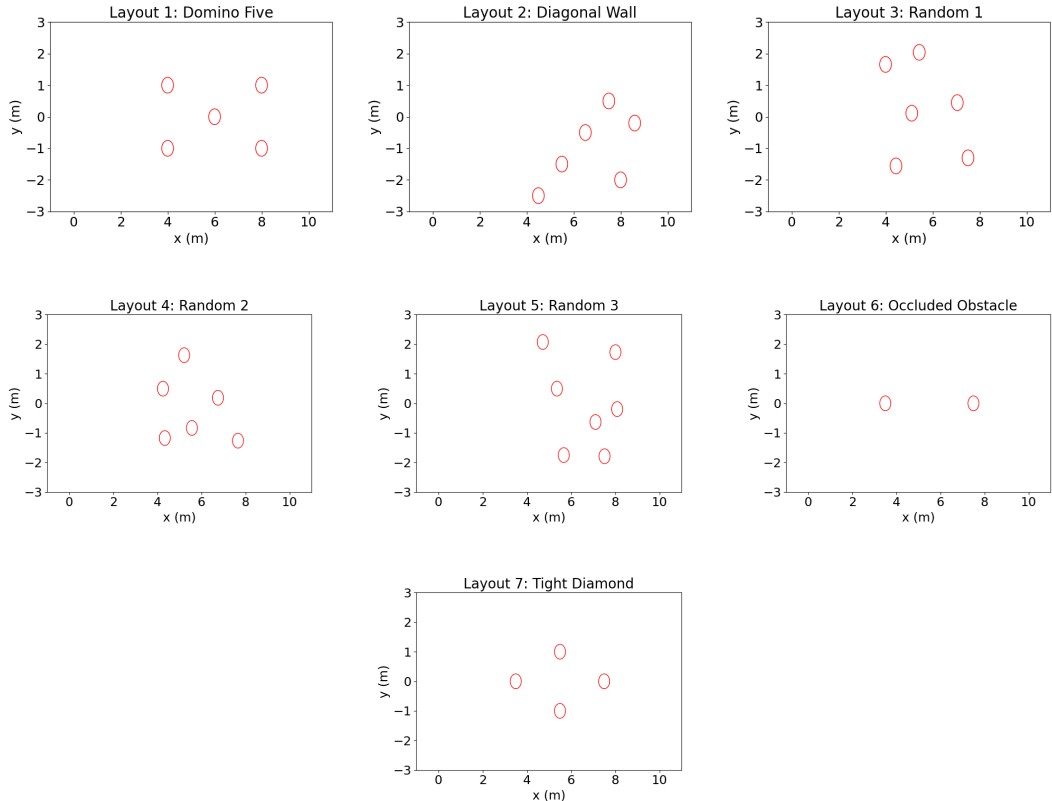

Figure 3: Obstacle layouts from a bird's eye view used during the experiments. Obstacles are denoted by the red circles. The quadruped was placed at (0, 0) and tasked with traversing 10m in the the x direction.

# D   Simulation Details and Resources

## D.1   Simulation Implementation: Parameters, Setup, and Runtime

In this section, we report the hyperparameters used for the experiments results in the main text. We implemented our algorithm and environments in JAX. All experiments were carried out on a single CPU in minutes.

We set the full horizon T to 100 and the history length H to 10. For random perturbation across environments, we sample noise from Gaussian distribution with mean 0 and standard deviation 0.5. For directional perturbation, we sample Gaussian noise with mean 0.5 and standard deviation 0.5. A random seed of 0 is used for all experiments. We obtain the nominal control from LQR with Q set to 0.001 and R set to 1. We then learn the residual obstacle-avoiding parameter M via gradient descent. The learning rate of gradient descent is 0.008 in the centerline environment and is 0.001 for the other environments.

An important note for these experiments: we do not implement existing heuristic techniques like obstacle padding to improve the RRT* collision-avoidance performance. As such, this performance is not meant to suggest that RRT* *cannot* work robustly in these settings, only that its nominal (and theoretically grounded) form does not account for disturbances or uncertainty and is therefore "optimistic" as compared to HJ methods, etc.

## D.2   Additional Figures and Trajectories – Centerline Environment

This appendix includes sample trajectories and other relevant visualizations for each algorithm.

### D.2.1   RRT* / A*

Here, we demonstrate some sample paths for RRT* in each disturbance regime. Fig. 4 shows uniform random noise, Fig. 5 shows sinusoidal noise, and Fig. 6 shows adversarial noise. In each case, the shift at the final time (goal position, top of image) causing a horizontal shift in the path should be ignored.

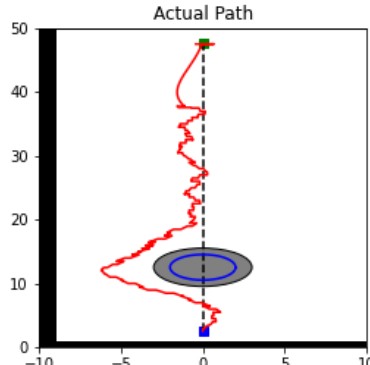 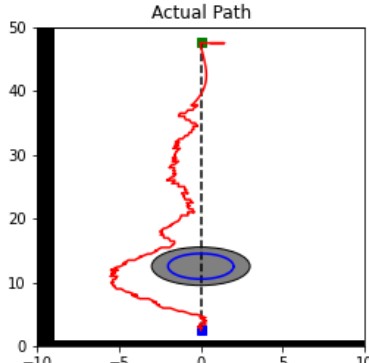

Figure 4: RRT* Planner trajectories against uniform random disturbances. Obstacle is the gray sphere, with the nominal trajectory a dashed black (vertical) line.

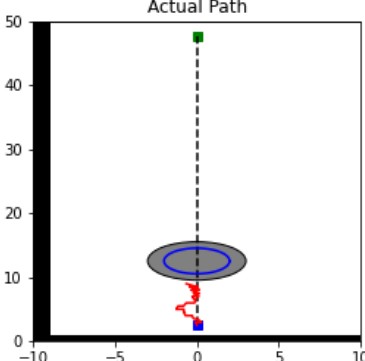 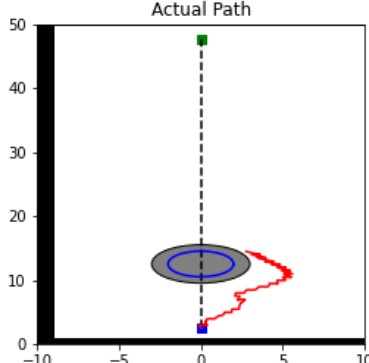

Figure 5: RRT* Planner trajectories against sinusoidal disturbances. Obstacle is the gray sphere, with the nominal trajectory a dashed black (vertical) line.

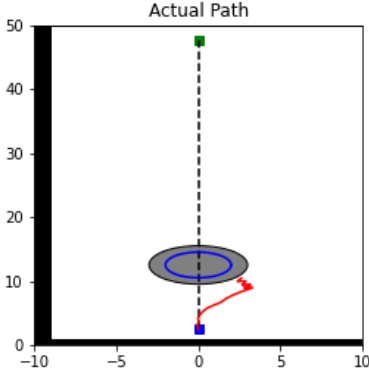 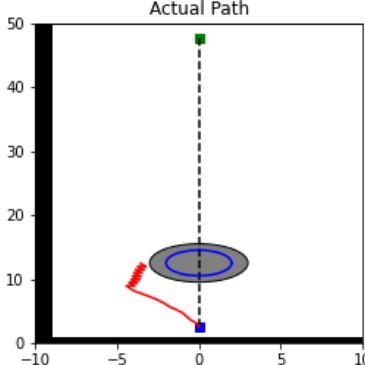

Figure 6: RRT* Planner trajectories against adversarial disturbances. Obstacle is the gray sphere, with the nominal trajectory a dashed black (vertical) line.

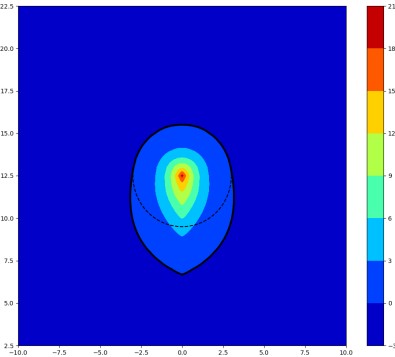

Figure 7: Racer backwards reachable set (inside thick black line) and the obstacle (dashed black line).

### D.2.2 HJ Reachability Planner

For the centerline example, the HJ Reachability planner constructs in Fig. 7 the backwards-reachable set for a given obstacle (dashed line), subject to the dynamics constraints imposed on the racer.

Note that every positive-value region denotes an unsafe region. The interpretation is that there is a "pseudo-cone" in front of the obstacle from which the vehicle cannot escape hitting the obstacle *if the disturbances are sufficiently adversarial*. Note that this means that HJ planning is independent of the *actual* disturbances. For each of the disturbance patterns (random, sinusoid, adversarial), we plot a collapsed view of sample trajectories around an obstacle for the HJ planner in Fig. 8. Note how similar each plot is, due to this independence of the control from the actual observed disturbances.

### D.2.3 Online Learned Planner

The key illustration here is that the trajectories of the online planner follow the structure of the disturbances, as illustrated by the following comparison of the uniform random and sinusoidal disturbances in Fig. 9 and Fig. 10.

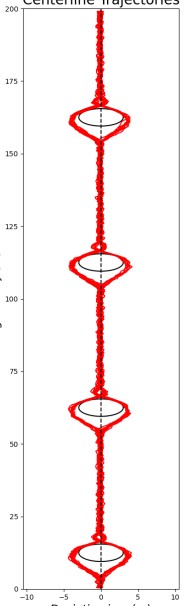 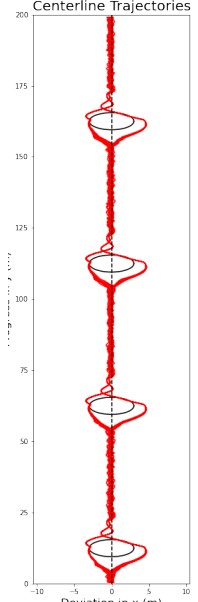 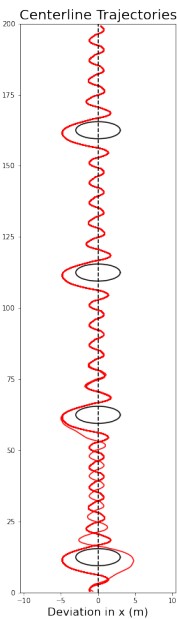

Figure 8: HJ Planner trajectories against (L) uniform random, (C) sinusoid, and (R) adversarial disturbances. Obstacles are black spheres, with the nominal trajectory a dashed black (vertical) line.

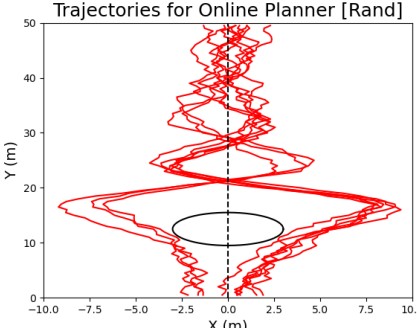 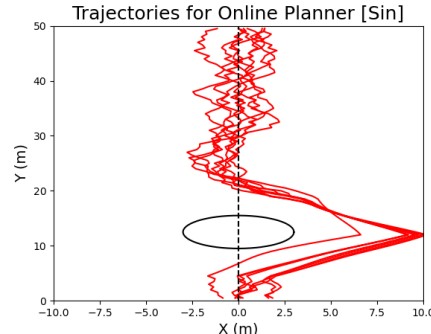

Figure 9: Collapsed trajectories of the racer using the online planner with random disturbances. The racer passes on each side evenly.

Figure 10: Collapsed trajectories of the racer using the online planner with sin disturbances. The racer learns to pass on the right.

### D.3 Experiment Details: Dynamic Pedestrian Environment

The pedestrian environment comprises many dynamic obstacles moving through the Racer scene, in which the Racer must avoid collision (contact). In this setting, the robot is not given direct obstacle state information, but must act instead on a history of observations of each (visible) obstacle's position *only*. The difficulty of the instances of this setting are illustrated in the following figures, which highlight several particular instances of success and collision respectively within the experiments conducted.

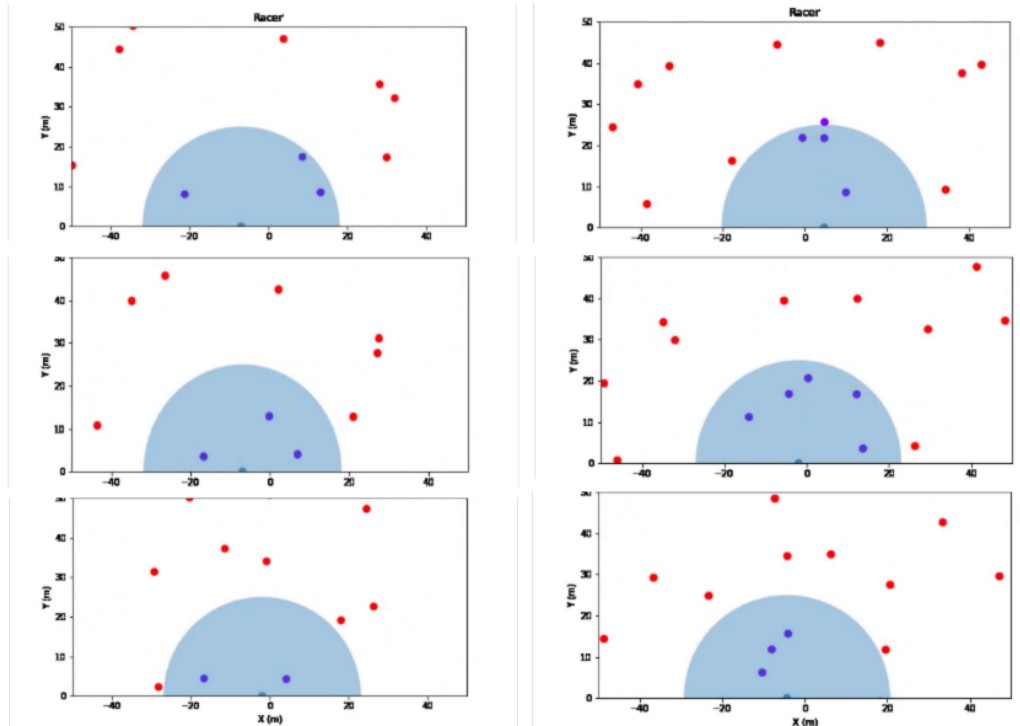

Figure 11: (L) Non-crash sequence of still images in which OLC successfully avoids a set of 4 close obstacles; (R) Crash sequence of images in which OLC cannot avoid a crash with one of 6 close obstacles.

We note several important details for the simulations. The pedestrians are non-avoiding, and in some cases move nearly as fast as the simulated Racer agent; the majority of collisions are observed due to a combination of (1) local density of pedestrians, (2) high relative speed of at least 2 pedestrians, and (3) non-avoiding nature of the pedestrian trajectories. These environments are more dynamic and chaotic than most in the related literature, as they are intended to demonstrate a 'proof-of-concept,' not to rigorously compare to the many available baselines (the comparison in static environments allows for more interpretable cases that highlight the relative advantages of stochastic (e.g., RRT), adversarial (e.g., HJ), and hybrid (e.g., OLC) methodologies, respectively). We think that adaptation to moving obstacles is very natural future work, and a possibly valuable point of future development for our algorithm.

### D.4 Failures in the Slalom Setting

We include an image of the four major environments in Fig. 12. Our simulated performance was strong in all environments except for the slalom environment, which we discuss further below.

The "slalom" setting allows us to tune its difficulty by varying the x-position (i.e. offset) of the gates or by narrowing their width. Fig. 13 illustrates the effect of increasing gate offset from center (i.e., error in the nominal planned trajectory – x-axis) and decreasing gate width (i.e., greater sensitivity to disturbances – y-axis) on failure rate through the slalom gates. As expected, reduced gate width

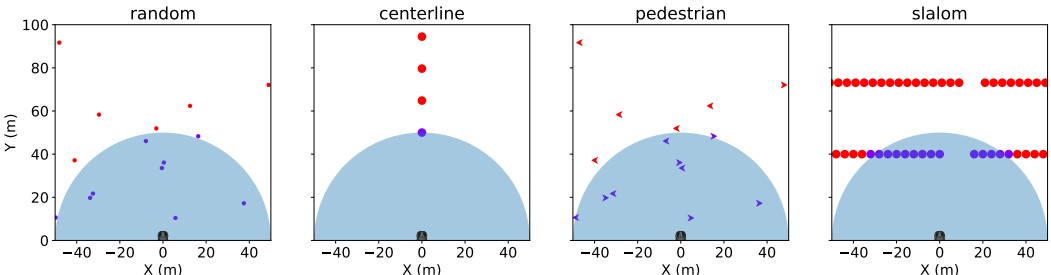

Figure 12: Illustration of the four environments used as a proof-of-concept for our Online algorithm.

and increased offset broadly increase failure rates. This is due to the online planner being forced to overcome a poor nominal planned trajectory; in combination with the gated passageways, this requires very precise sequences of inputs and a longer memory of previously observed gates (due to the limited sensing horizon). This is discussed further in Supp. D.4.1.

### D.4.1 Discussion

The first answer is relatively direct: in all of our examples, we are implicitly acting in a kind of Frenet frame, where all obstacle positions and other referencing is to the ego vehicle (racer) position. As such, the nominal planned trajectory can always be thought of as mapped to a straight line ahead of the racer. In this context, some slalom gates represent a 20m deviation *from the nominal trajectory.* However, this flies in the face of the central modeling intuition of the online framework – that obstacle avoidance is local, with local sensing, local deviations from the nominal trajectory, and "reactive" control to disturbances as they arise. In this vein, the nominal slalom is a challenging task, precisely because it stretches the limits of what can be met by our setup. Concretely: limited sensing makes each slalom wall a kind of "gradient-less" observation (shifting left and right yields only a continuation of the wall *unless the gap is already sensed*), meaning that choosing the correct Left/Right action is difficult. Additionally, the map displays memory, because going the wrong way early through one gate can render the next gate infeasible.

It is in light of these considerations that we argue that the slalom case is actually a case for our model, because it interpretably creates a setting in which the key assumptions are broken. Just like an actual skier who overshoots through one gate and cannot recover for the next gate, so too does our obstacle avoidance algorithm run the risk of "dooming" itself due to a wrong turn – but this is, as described, fundamental to the hardness of the obstacle avoidance problem! As such, we consider the slalom gate as a fundamentally hard problem, and consider a case for future work a fuller characterization of how our planner works for slaloms of varying difficulty, as measured by the sensor range, the distance between gates (both laterally and longitudinally), and the fundamental "cost memory" as it depends on these and other parameters.

### D.4.2 Experimental Parameter Sweep – Slalom Course

In an effort to better represent the effects of the slalom setting and the dependence on gate width and offset, Fig. 13 illustrates the effect of increasing gate offset from center (i.e., error in the nominal planner trajectory – x-axis) and decreasing gate width (i.e., greater sensitivity to disturbances – y-axis) on failure rate through the slalom gates. As expected, reduced gate width and increased offset broadly increase failure rates.

We note that for narrow gates, a zero-offset slalom is actually quite challenging to ensure - we believe this is due to the fundamental $H_\infty$ limit of stabilization of disturbances for this system; namely, a too-narrow gate requires too-strong robustness about the setpoint (origin), causing failure. This also explains why failure rates are high but not one for moderate offsets in the narrow-gate environment as well: specifically, they allow some freedom away from the zero-offset regularization problem.

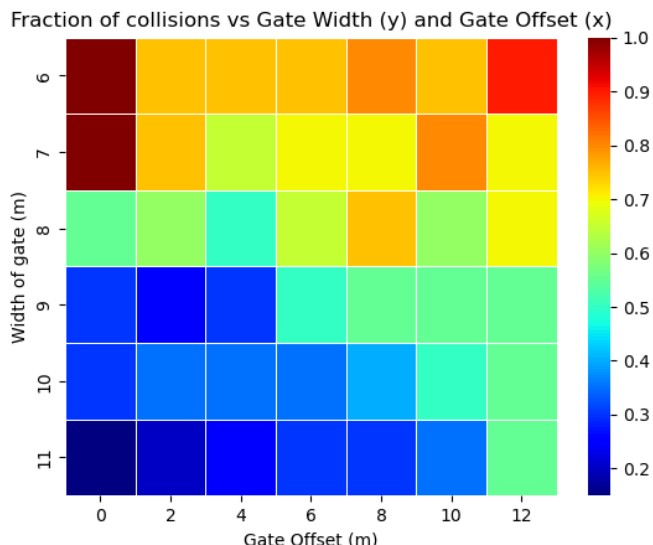

Figure 13: Failure rate heatmap for increasing gate offset (left to right) and decreasing gate width (bottom to top). As expected, failure increases with narrower gates (top) and larger offsets (right).

Besides this, the trend quite clearly demonstrates the fundamental increases in difficulty observed for narrower gates and larger offsets, as expected.

