# OpenReview forum: "Online Learning for Obstacle Avoidance"
_robot-learning.org/CoRL/2023/Conference — CoRL 2023 Poster_

### Official Review · Reviewer_63fg · 2023-07-14

**Confidence:** 4
**Originality:** Good
**Technical Quality:** Good
**Clarity Of Presentation:** Good
**Impact:** 3

**Recommendation:**

Weak Accept: I recommend accepting the paper, but will not argue for my recommendation if the majority of other reviewers have a different opinion.

**Review:**

The topic addressed is significant and very relevant to robotics and the proposed algorithm seems promising.
The paper adapts an algorithm combining previous works ([38]+[52]) for obstacle avoidance. Therefore, the theoretical contribution needs to be clarified here - this is also due to the fact that it is not explained what exactly these previous works do.

The related-work part does not seem complete; there are more works than ones that probabilistic model - based and worst-behaviour-based. For example, the adaptive control works by Dimarogonas group:

Verginis and Dimarogonas. "Adaptive robot navigation with collision avoidance subject to 2nd-order uncertain dynamics." Automatica, 2021.
Verginis, Dimarogonas, and  Kavraki. "Kdf: Kinodynamic motion planning via geometric sampling-based algorithms and funnel control." IEEE Transactions on robotics, 2022.

that adapt online to uncertainties and disturbances.

The proposed algorithm further needs some clarification. For instance, what exactly tilde{M} is and how are the corresponding x and u (with tilde{M} superscripts) rigorously defined. Is the Initialize procedure in Alg. 1 executed offline in some simulation environment? Because it requires runs of the system to observe the several properties.

Similarly, some more details throughout the paper can be provided, such as how is (4) exactly derived from (1)-(3), what exactly is the FPL methods, or what [38,52] consider.

References to the supplementary material should be avoided (like see Supp. B in Theorem 2) and the picture references in the experiments.

I understand the space limit (also with respect to my comment on more clarifications) and that a lot of information is in the appendices, but the paper needs to be compact in that sense.










**Quality Of The Limitations Section:**

Limitations are addressed clearly

**Questions For Rebuttal:**

- Please clarify the theoretical contribution wrt previous works, especially [38,52]
- Please include adaptive-control works in the related literature, like the ones suggested.
- Please clarify details of the proposed algorithm as per my comments above


**Robotics Focus:**

Sufficient demonstration on hardware

**Summary Of Paper:**

The paper presents an algorithm for robot navigation with obstacle avoidance.
The algorithm is robust against uncertainties and perturbations and adapts online, also to local obstacle sensing.
Theoretical results are provided and extensive simulation and experimental results are conducted.

**Summary Of Recommendation:**

I believe the paper presents a strong algorithm for navigation and collision avoidance, but modifications are needed based on contribution, literature, and clarity.

---

> ### Author Response · Authors · 2023-08-08
> **Response to Reviewer 63fg (Part 1 of 2)**
>
> Thank you for your detailed comments and clarifying questions. We respond to five main points (numbered 1-5).
>
> **(1) [Clarify Theoretical contributions] -- also see (4), below]**
>
> The overall setup of our contribution is also outlined in the response to Reviewer DrfS’s question
> concerning the relationship between Eqn. 9, trust region problems, and Follow-the-Perturbed-Leader
> algorithms. We outline the theoretical contributions as follows:
> * Specifying an Instance of Equation 8 (Overcoming the $\min$ Operator)
>     1. Proving that the problem specification is the equilibrium solution of a specific two-
> player general-sum game.
>     2. Demonstrating that the optimal strategy is a trust region instance for the ‘protagonist’
> for any game state, and that the game as a whole is convex-concave. Further, we need
> to show that each player has a $\tilde{\mathcal{O}}(√T )$-regret (near-optimal) algorithm to play.
>     3. Using [61] to guarantee the convergence and regret properties of this ‘specification
> game’.
> * Extension to the Multi-Instance Case [Equation 9]
>     1. Formalizing the multi-step case as a ‘lifted’ (i.e., larger) game, with proper interpretation (i.e., the ‘antagonist’ can optimize over each instance independently).
>     2. Demonstrating that the resulting game remains convex-concave, with low regret as in
> the above setting. This means that the optimization oracle needed for FPL is feasible.
>     3. Using [38] and [52] (modifying some constants and following through with some
> domain-specific calculations) to establish that our problem is an FPL (with Memory)
> instance and attains the regret claimed in Theorem 2.
>
> We will emphasize these technical contributions further in the revision.
>
> **(2) [Related Work]**
>
> We thank the reviewer for pointing us to these references. We believe that, for the specific examples
> given, the results cited by the reviewer lie in a ‘worst-case’-type formalism, both in terms of the
> structure (bounded, continuous-time disturbances), and in terms of the resulting guarantee. To further
> this claim, we note that, in the examples given, the controller is proven to be globally stable, which
> amounts to a kind of ‘uniform disturbance rejection;’ it does not, to our interpretation, suggest that
> the controller is learning to strongly reject the specific type of disturbance (e.g., sinusoids, constants,
> etc), but rather is capable of (at least weakly) rejecting _all_ types. We will include several additional references in this area, as they demonstrate an alternative formulation for disturbance rejection in the
> obstacle avoidance context.
>
> **(3) [Algorithm Detail Clarifications]**
>
> The $\tilde{M}$ ambiguity was an imprecision in the notation.  $\tilde{M}$ is a dummy (optimization) variable. The draft combined
> two ideas in Eqn. 8, which should have been separated. To wit, Eqn. 8 should be in $\tilde{M}$, where the
> _particular realized reward_ is an evaluation at the particular $\tilde{M}\_t$. Currently, it mixes parts of each in a
> confusing fashion.
>
> The specification of $x\_{t+1}$ and $u\_{t}$ are implicitly defined by  $\tilde{M}$, $x\_{t}$, and {$w\_τ$}$
> \_{τ =1}^t$,
> where a particular $x\_{t+1}$ is calculated via Eqn. 4, using a $u\_t$ calculated as in Eqn. 3.
>
> For the initialization, all of the parameters are either initialized randomly ($P\_0$, possibly $x\_0$), or are
> derivable a priori from the analysis of the regret bounds (i.e., exponential parameter η, regularization
> scalar λ). There is a typo, which is that the input obstacles should have a subscript of ‘0,’ meaning
> only the ‘initial sensing input;’ they are updated in Algorithm 1 at each time step within the **for** loop.
>
> As to the initial random play: the horizon length $H$ is set (in our case) to a short period of 5 time
> steps, making the requirement less onerous. Furthermore, the algorithm need not play randomly at the
> outset, but can initialize the disturbance history for t < 0 to be uniformly zero, and play accordingly.
> The random play is an artifact from the regret analysis to specify that ‘no initial learning stage as
> short as $H$ can significantly affect the regret metric.’ The length of this period is about 1 second; we
> could assume that in this time the quadruped simply remains still.

---

> > ### Comment · Reviewer_63fg · 2023-08-10
> > **some further algorithm clarifications**
> >
> > Thanks for the reply.
> >
> > In the revised version, it should be explained more clearly how eqn. (8) and (9) are computed/implemented.
> > Do you create, in (8), a function of $\tilde{M}$ to be optimized in (9) with respect to this $\tilde{M}$ (or this dummy variable-argument)? In this case, what does 'counterfactual' refer to?
> > Or are there specific $\tilde{M}$ used in (8) to calculate the trajectories (other than the ones used in (3))?
> >
> > Additionally, I believe that some terms, like $d_u$ and $d_w$ are not defined (although it seems they are the dimensions of u and w).
> > Finally, what is the purpose of using P_0? Since the disturbances are calculated in subsequent steps and taken into account in the algorithm.

---

> > > ### Author Response · Authors · 2023-08-10
> > > **Re: Further algorithm clarifications**
> > >
> > > The former statement is _close_ to the methodology. For each $t$, we create in (8) a function of $\tilde{M}$ (the set of all possible policies in our class) that reflects the current `context' of the problem (the current relative obstacle positions, current disturbance history, current state). Note that $b$ can be incorporated into $\tilde{M}$ by appending a value of $1$ to the disturbance measurement. We then show that (8) is a trust region instance.
> > >
> > > All instances of (8) are then summed for $\tau \in ${$1, ..., t$} in (9); this summation is still a trust region instance, and the resulting function is still in $\tilde{M}$ (still in the same class of policies). This exactly matches the Reviewer's statement, except that it is not (8) _directly_ that is optimized in (9), but rather the _summation of all (8)'s up to the current time step_ (this may have been the intended meaning, we just want to ensure clarity that it is not Eqn. (8) _directly_). Then the reviewer's above statement is again completely correct. Specifically, in (9) we are optimizing over the dummy variable $\tilde{M}$; the counterfactual set is then set of all paths induced by the choice of alternative $\tilde{M}'$ from the feasible set. Just as we needed to remove the $t$-subscript in Eqn. 8, we need to remove it from the discussion of the counterfactual set in Section 4.3. The counterfactual set is all feasible $\tilde{M}$.
> > >
> > > We have updated the manuscript to clarify in Section 3 that $\bar{\mathbf{x}} \in \mathbb{R}^{d\_x}$, $\bar{\mathbf{u}} \in \mathbb{R}^{d\_u}$, and $\bar{\mathbf{w}} \in \mathbb{R}^{d\_w}$. In the presentation given $d\_x = d\_w$, but one can add wlog a disturbance-to-state matrix $G$ and allow for $d\_w < d\_x$ without affecting our results. (In that case, the state transition is $x\_{t+1} = A\_0x\_t + Bu\_t + Gw\_t = \tilde{A}x\_t + B\tilde{u}\_t + Gw\_t$). The manuscript presents the case $G = I_{d_x}$, which affords the disturbances the greatest amount of `power.' We will add this commentary in a footnote.
> > >
> > > The purpose of using $P\_0$ is to form the regularizer (the 'perturbation'). If we did not include it, we would optimizing solely over the best policy (a particular $\tilde{M}\_t^*$) in hindsight through time $t$. This (intuitive) algorithm would be 'follow-the-leader' (FTL), _not_ 'follow-the-perturbed-leader' (FPL).  However, as intuitive as the FTL algorithm is, it _does not_ have sublinear regret, so cannot be used to obtain a guarantee. However, the property "is a trust region instance in $\tilde{M}$" is closed under addition of linear terms in $\tilde{M}$, so we can add the linear perturbation and still have a trust region instance. It is important to note that the perturbation structure is _not_ arbitrary, but actually the result of careful analysis (in other works); that is, the specific, theoretically-grounded perturbation structure happens to be a linear perturbation, which means that our necessary property (the trust region instance in $\tilde{M}$) is preserved.

---

> > > > ### Comment · Reviewer_63fg · 2023-08-11
> > > > **Thank you note**
> > > >
> > > > Thanks for the reply, the algorithm is clearer now; please try to add some of these clarifications in the revised version.

---

> ### Author Response · Authors · 2023-08-08
> **Response to Reviewer 63fg (Part 2 of 2)**
>
> **(4) [Equation 4, FPL Methods, context of [38], [52], [61]]**
>
> We will explain these points more clearly in the revision. Regarding Eqn 4, we move the $Kx\_t$ out of the $u\_t$ expression
> and into a ‘closed-loop’ $A$ matrix (just as one would do to convert a nonautonomous system into an
> autonomous one using state feedback). To include the details for completeness:
>
> $x\_{t+1} = Ax\_t + Bu\_t + w\_t$
>
> $u\_t = Kx\_t + b\_t + [f (M, w)]$
>
> $\implies x\_{t+1} = Ax\_t + BKx\_t + B(b\_t + [f (M, w)]) + w\_t$
>
> $= (A + BK)x\_t + B(b\_t + [f (M, w)]) + w\_t$
>
> $=  \tilde{A}x\_t + B\tilde{u}\_t + w\_t$
>
> We will add a comment in the revision to clarify the precise nature of this transformation. As a
> side note, this whole transformation acts to ensure that our ‘zero-algorithm-input state transition’
> (i.e., post-state feedback but pre-online adaptation) matrix  $\tilde{A}$ is a stable matrix. Implicit in this
> transformation is the necessary (and intuitive) condition that the system is stabilizable.
>
> We will update the draft to include a brief description of the application tackled in [38] and describe
> more clearly our theoretical contributions within the context of [38], [52], and [61]. The following
> paragraph explains the key points we will include:
>
> [38] develops a method to generate adversarial disturbances for linear systems online, and extends
> the FPL algorithm to the setting with memory, where it is used to prove a regret bound for the
> disturbance generation problem. [52] demonstrates that FPL is near-optimal in the non-convex setting
> if an optimization oracle is present; the result also holds for the convex case (many optimization
> schema function as ‘oracles’ in the easier convex setting). [38] shows that the FPL with memory
> result inherits this near-optimality shown in [52]. Finally, [61] gives regret bounds for many game
> scenarios in which the players have varying cost landscapes. From this work, we specifically use the
> results for convex-concave games.
>
> **(5) [References to the Supplement]**
>
> The current presentation was intended to provide additional ‘scaffolding’ for the reader to more
> quickly find relevant details as to how the various theoretical/technical components fit together. For
> any final publication, these will be excised or reformatted to conform to the conventions of the venue.

---

> > ### Comment · Reviewer_63fg · 2023-08-10
> > **Update of $\tilde{u}_t$**
> >
> > Thanks for the reply.
> > The revised version should be updated with the definition of $\tilde{u}_t$ according to the aforementioned analysis.

---

### Official Review · Reviewer_x84o · 2023-07-17

**Confidence:** 4
**Originality:** Good
**Technical Quality:** Good
**Clarity Of Presentation:** Good
**Impact:** 1

**Recommendation:**

Weak Reject: I recommend rejecting the paper, but will not argue for my recommendation if the majority of other reviewers have a different opinion.

**Review:**

Strengths: The paper makes a novel technical contribution in that it a. poses the obstacle avoidance problem in a regret minimization framework and b. develops a trust region-based online learning algorithm for obstacle avoidance with provable bounds. The paper is clearly written and easy to understand.

Weaknesses: Modern robots have many failings. However, running into things is not high on the list. When we live in an era of driverless cars that have driven on public roads for millions of miles without a collision, it is questionable whether stationary obstacle avoidance is a significant enough problem to warrant an attack. I am (largely) on board with this paper in terms of its intellectual contributions, but I am not convinced it is solving a significant or important problem. A further limitation is that the comparisons are made with a narrow class of baselines and only in a setting where the obstacles are stationary.


**Quality Of The Limitations Section:**

Limitations are addressed clearly

**Questions For Rebuttal:**

Since my main negative comment about the work is its significance, I'd appreciate hearing why the authors think pursuing this problem is worthwhile. A second, less important issue is the comparison with existing methods. Most obstacle avoidance on real systems is never done open loop. Can the authors provide a comparison with 1. a reactive system and 2. a setting in which obstacles are dynamic?


**Robotics Focus:**

Relevant but unlikely to deploy to hardware in near future

**Summary Of Paper:**

The paper frames the robot obstacle avoidance problem from the viewpoint of online learning. The policy adapts online to uncertainty and is shown in hindsight to be provably comparable to the best obstacle avoidance policy (from a restricted class). The method is validated in simulation and compares favorably to baselines (open loop planning and Hamilton-Jacobi reachability). It is also implemented on a hardware example.

**Summary Of Recommendation:**

This paper has some nice aspects but is not significant enough to be published at CoRL.

---

> ### Author Response · Authors · 2023-08-08
> **Response to Reviewer x84o (Part 1 of 2)**
>
> Thank you for your detailed comments and clarifying questions. We respond to the two questions given in the Questions for Rebuttal. All cited works are included at the end.
>
> **(1) [Why Obstacle Avoidance]**
>
> We thank the reviewer for raising this question and appreciate the opportunity to clarify the significance of our contributions.
>
> First, we note that obstacle avoidance writ large is important for applications beyond autonomous
> driving [2 , 3]. In addition to other navigation and exploration problems, there are applications to
> manipulation and other quasi-static settings (e.g., [4 , 5, 6]). Furthermore, our method is not restricted
> to stationary objects, which increases the range of potential applications. In other words, stationary
> obstacle avoidance is still an interesting problem, and our algorithm can also be applied in dynamic
> contexts.
>
> Second, we can use the example of autonomous driving to motivate the development of new theoretical
> paradigms: despite the significant application of capital, the significant efforts of many engineers,
> the enormous amounts of training data, the capacity for lots of onboard computation and rich sensor
> suites, and the generally safe empirical record, relatively few people would argue that we have
> a definitive framework to confidently certify that the policies are in fact safe, beyond citing the
> empirical performance so far. One could foresee a point at which the empirical record becomes so
> overwhelmingly strong as to obviate this need, but such a point is exceedingly costly and inefficient
> to reach - and we should not like to have to repeat it for each new application domain. Developing frameworks both for synthesizing safe policies and for certifying policy safety is very important both for safety itself and for confident certification of methods that don't (or can't) explicitly consider safety in the policy synthesis. To this end, our work contributes a novel regret minimization formulation and algorithm for obstacle avoidance (potentially in the presence of dynamic obstacles), which provides a certificate of reliable performance
> which can complement empirical evaluations.
>
> To expand on the broader context, a major strength of online algorithms is that they are adaptive in
> such a way as to be ‘instance-optimal,’ meaning that they approach optimal behavior for whatever the
> realization of uncertainty – be it benign (the controller will evolve to become more optimistic) or
> adversarial (the controller will become more cautious), or somewhere in between. Such formalisms can find a middle ground between the tight (optimistic) margins of methods like A* and the loose
> (conservative) margins of methods like HJ reachability or those in the suggested resources of Reviewer
> 63fg, when such gaps exist.
>
> We conclude by noting that both of the other reviewers highlighted the significance of the problem
> we consider in their reviews.
>
> **(2) [Closed-Loop vs. Open-Loop; Dynamic Obstacles]**
>
> We want to make an important clarification regarding the terms ‘open-loop’ and ‘closed-loop’ as used
> in our paper. The draft will be revised to ensure that the following points are more clearly expressed
> in the text.
>
> The setup for every algorithm includes the following:
> * A high-level planner (A*, RRT*) generating a nominal plan consisting of waypoints (a path)
> from the current position to the goal, and open-loop (nominal) inputs to reach each of those
> waypoints. This path is generated with knowledge of the robot’s current observations of
> obstacle locations.
> * Under the hood (something we do not touch) is a joint-level controller choosing arm/leg
> movements to effect the desired higher-level behavior (e.g., forward motion, right or left
> turns). This operates very fast (> 100Hz).
> * A mid-level path-following controller (that we created) providing feedback to ensure that
> the robot tracks the waypoints. This is the ‘interior’ layer of feedback, operating at ≈ 5-10
> Hz.
> * The high-level planner replanning at ≈ 1 Hz, which serves as feedback that incorporates
> new information – newly sensed obstacles, current position estimates, etc.
> * (OLC-Specific) The mid-level path-following controller referenced above is augmented with
> an OLC module providing additional ‘safety correction inputs’ in between replanning steps.
>
> As such, every method implemented is closed-loop on two levels (mid- and high-level, in our
> nomenclature). When we referred to open-loop or ‘nominal’ paths, we meant only to refer to ‘the
> current plan’ (which is a set of open-loop inputs and the projected waypoints, that will be subsequently
> updated in a receding horizon manner). As noted before, we will clarify this architecture in the
> revision.

---

> ### Author Response · Authors · 2023-08-08
> **Response to Reviewer x84o (Part 2 of 2)**
>
> (2) [Continued]
>
> As to the question regarding dynamic obstacles: nothing in our algorithm precludes the inclusion of
> either (1) explicit additional state information of the observed obstacles (velocities) or (2) a history of
> obstacle locations (from which each obstacle’s velocity may be indirectly inferred). In support of this,
> we have simulation results for our algorithm operating in a dense moving-pedestrian environment,
> which were not included due to space constraints (they were only mentioned briefly in Supp. D).
> These results were designed to see if the algorithm could successfully avoid obstacles and did not
> compare to RRT* / A* explicitly. We can run additional simulations in these settings to give additional
> insight into the performance generalization to dynamic obstacles, as well as incorporate an A* or
> RRT* baseline. We will emphasize the ability to handle dynamic obstacles in the revision and will
> add a description of the corresponding experiments to the revision.
>
> References:
>
>
> [2] P. Lasota, T. Song, and J. Shah. A Survey of Methods for Safe Human-Robot Interaction. URL
> https://ieeexplore.ieee.org/document/8186877/.
>
> [3] G. Du, S. Long, F. Li, and X. Huang. Active Collision Avoidance for Human-Robot Interaction
> With UKF, Expert System, and Artificial Potential Field Method. Frontiers in Robotics and
> AI, 5, 2018. ISSN 2296-9144. URL https://www.frontiersin.org/articles/10.3389/
> frobt.2018.00125.
>
> [4] A. Sepehri and A. M. Moghaddam. A Motion Planning Algorithm for Redundant Manipulators
> Using Rapidly Exploring Randomized Trees and Artificial Potential Fields. IEEE Access, 9:
> 26059–26070, 2021. ISSN 2169-3536. doi:10.1109/ACCESS.2021.3056397.
>
> [5] J. Pankert and M. Hutter. Perceptive Model Predictive Control for Continuous Mobile Manipu-
> lation. IEEE Robotics and Automation Letters, 5(4):6177–6184, Oct. 2020. ISSN 2377-3766.
> doi:10.1109/LRA.2020.3010721.
>
> [6] M. C. Yip and D. B. Camarillo. Model-Less Feedback Control of Continuum Manipulators in
> Constrained Environments. IEEE Transactions on Robotics, 30(4):880–889, Aug. 2014. ISSN
> 1941-0468. doi:10.1109/TRO.2014.2309194.

---

> ### Author Response · Authors · 2023-08-11
> **Update and Additional Clarifications**
>
> Dear Reviewer, we would appreciate it if you could let us know if our rebuttal has addressed the points raised in your review. We are happy to engage in further discussions. Thank you again for your time and effort in providing feedback on the paper.

---

> > ### Author Response · Authors · 2023-08-14
> > **Follow-Up for Additional Clarification**
> >
> > Dear Reviewer, we would appreciate it if you could let us know if our responses have addressed the points raised in your review. We are happy to engage in further discussions. Thank you again for your time and effort in providing feedback on the paper.

---

> > ### Comment · Reviewer_x84o · 2023-08-14
> > **Thanks for your response.**
> >
> > Thanks very much for your response. I'll leave it up to the area chairs and program committee to decide on the paper's significance when they make the decision on whether to include your paper in the program.

---

### Official Review · Reviewer_DrfS · 2023-07-20

**Confidence:** 3
**Originality:** Very Good
**Technical Quality:** Very Good
**Clarity Of Presentation:** Good
**Impact:** 4

**Recommendation:**

Weak Accept: I recommend accepting the paper, but will not argue for my recommendation if the majority of other reviewers have a different opinion.

**Review:**

Review Summary
Overall the problem is well-motivated, and the problem well set up. There are a couple of unclear sections in the technical presentation (see below), but overall the approach seems both correct and reasonable. The empirical results are good.

Detailed Comments
* The symbol $A$ seems to be overloaded: in eq. 2, it is the controller, and in eq.1, it is the state transition matrix. Is this correct?
* In Section 4.3, it is unclear what the connection between the optimization in eq. 9 and FPL is. It seems to just be a trust-region (potentially nonlinear, non-convex) optimization problem. What algorithm is used? If a local optimization solver is used, how is the solution initialized?
* It seems that the horizon $H$ should be carefully chosen to be long enough to allow for observability of unmodelled dynamics at all times, while minimizing it for computational tractability. Is this true? How is it chosen?
* The switching of planning algorithms between sim and real is odd - why not try both RRT* and A* in both sim and real, as baselines?
* The robustness of both A* and RRT* is dependent on the margin, as identified in the paper. However, this can affect the planner's ability to find narrow paths - it would be nice to highlight this empirically and compare it to the proposed approach.

**Quality Of The Limitations Section:**

Limitations are addressed clearly

**Questions For Rebuttal:**

Please respond to the questions under the detailed comments above.
* How sensitive is the algorithm to accurate estimation of the disturbances $w_t$?


**Robotics Focus:**

Sufficient demonstration on hardware

**Summary Of Paper:**

This paper presents an online learning approach to robust obstacle avoidance in the presence of unmodelled system dynamics and environmental perturbations. The proposed algorithm performs online trust-region optimization of a linear dynamic controller using an obstacle margin loss regularized by perturbations to the nominal plan, to ensure robustness while straying as little as possible from the nominal plan.

**Summary Of Recommendation:**

Overall this seems like a strong paper, though there is room for improvement in the technical presentation, and in the explanation of the algorithm. If they can be resolved satisfactorily, I would be happy to revise my recommendation.

---

> ### Author Response · Authors · 2023-08-08
> **Response to Reviewer DrfS (Part 1 of 2)**
>
> Thank you for your detailed comments and clarifying questions. We respond to the five bullet points (responses numbered 1-5) as well as the rebuttal question (numbered 6). Any cited works are included in the last post.
>
> (1) Yes, there was an overload in the term $A$; we have updated the draft to denote the dynamics term
> (Eqn. 1 and elsewhere) as $A_0$ for clarity.
>
> (2) This is a very good question, as it is fundamental to understanding how OLC works. We clarify the
> interplay between Eqn. 8, Eqn. 9, trust region problems, and FPL below. Additionally (per Reviewer
> 63fg’s comments), we have updated the draft to give some background and discussion of sources
> [38], [52], and [61], in order to clarify how our theoretical treatment builds upon their results.
>
> Eqn. 8 is the instantaneous reward. It is a trust region instance, but _cannot be explicitly specified a
> priori_ due to the $\min$ operator. The specification must instead be found as the limiting equilibrium of
> a convex-concave game (which is efficiently solvable from [61]).
>
> From this, Eqn. 9 combines a random perturbation with all of the preceding instances of Eqn. 8 up
> through time $t$ (that is, all of the past decisions and their ‘contexts’ – i.e., relative obstacle positions,
> state deviations, observed disturbances). The resulting policy ($M$) is selected as the $M$ that does best
> in aggregate over these decisions (the ‘leading’ $M$). Hence, solving Eqn. 9 amounts to choosing the
> ‘perturbed leading policy.’ Therefore, we are ‘following the (perturbed) leader’ [so far, through time $t$].
> FPL is sometimes referred to as a ‘meta-algorithm’ in order to distinguish it from optimization
> algorithms like (stochastic) gradient descent or numerical integration methods. Concretely, we will
> need to use an optimization algorithm as a subroutine of the FPL ‘meta-algorithm’ in order to
> iteratively compute the leading policy.
>
> FPL is known to be near-optimal, assuming it can actually be implemented (this is shown in [52]).
> The technical challenge, then, is making sure that one can actually solve for the leading policy at
> each time step. Therefore, we must ensure that the combination of all of these trust region instances
> (all of the instances of Eqn. 8 through time $t$) remains a trust region instance. The primary concern is ensuring
> that the $\min$ operators for each past instance can be overcome in parallel while retaining the property
> that the resulting ‘combined’ convex-concave game remains efficiently solvable (it does).
>
> To actually implement the optimization step, a trust region solver can be used in principle (and was
> used in our preliminary simulations). However, due to the underlying convexity of the problem and
> the nature of the adversary in the convex-concave game (which uses exponentiated gradient in their
> updates), it is actually faster and more stable (time-wise) to use stochastic gradient ascent on the
> rewards. Initialization of $\tilde{M}\_{0}$ is random, and for future time steps the preceding solution  $\tilde{M}\_{t−1}$ is
> used as the initialization to find $\tilde{M}\_{t}$. The intuition for stochastic gradient methods is that only the
> correct solution is a stable stationary point; all of the other stationary points are unstable. There is a
> significant body of literature on the behavior of optimization methods around saddle points; most
> relevantly, we find that in practice the stochastic gradient algorithm converges quickly, which is
> consistent with the results shown for perturbed gradient methods [1].
>
> (3) The horizon term is important to tune, for precisely the reasons given by the reviewer; it trades off
> state estimate accuracy (as used in the regret proof) with computational speed. In Supplement A,
> we show that, akin to Ghai, et. al. [38], the necessary minimum size of $H$ to get the stated regret
> is $\mathcal{O}(\log T)$. The important takeaway from this is that $H$ can be quite small. This means that small
> perturbations don’t persist too long in the system, which is a consequence of the assumed linearity
> in the dynamics. In our experiments, we used a small horizon length of $H = 5$ time steps. For
> simulations (which had larger $T$), we used $H = 10$ time steps.
>
> (4) The switch to A* for hardware experiments was undertaken solely to yield better path following for
> the quadruped robot we use (which we model and control as a Dubins’ Car). The implementation of
> RRT* that we used in simulation resulted in paths that still contained some sharp corners and turns,
> which would persist post-filtering to a sufficient extent as to degrade path following on hardware. The
> sharp corners and turns persisted in part due to the non-holonomic dynamics model and due to the
> time delay in the 4Hz update frequency. A* worked more repeatably in these regards.

---

> ### Author Response · Authors · 2023-08-08
> **Response to Reviewer DrfS (Part 2 of 2)**
>
> (4) [Continued]
>
> We justify the argument for A* not being a ‘significant’ change on hardware for the following reasons:
>
> 1. Both RRT* and A* are disturbance-agnostic, and incorporate margin in essentially the same
> way.
> 2. A* was discretized densely (10 cm x 10 cm resolution), so it should function close-to-optimally (behave similarly to RRT*).
> 3. Because our hardware obstacles were cylindrical, we could explicitly verify that A* would
> not connect two nodes in the free space that required the intervening path to intersect an
> obstacle.
> 4. Following on from the previous point, because A* cannot ε-approach (‘hug’) an arbitrary
> obstacle the way that RRT* paths can, it should actually be more robust than RRT* in our
> context, which reduce OLC’s apparent benefits.
>
> (5) The finding of narrow paths is an interesting point, but would not demonstrate the benefit of our
> algorithm as we primarily envision it, which is to select wider (‘safer’) paths than RRT* or A*.
> However, it would demonstrate an illustrative counterfactual. Namely, it would show that using
> a high-margin RRT* or A* fails to work as well, because instead of finding a good path (with lower-margin A*/RRT*) and following it robustly (using OLC), the high-margin A* / RRT* case
> would take a very circuitous path instead, essentially reducing to HJ methods.
>
> (6) The algorithm we propose demonstrates empirical robustness to imprecise disturbance reconstruction.
> This can be seen in the hardware experiments, where disturbances are computed based on noisy
> state estimates. Specifically, because disturbances are estimated based on the mismatch between
> the state predicted by the actual model and the realized state, any state estimation error is passed
> through to the disturbance reconstruction. This finding is consistent with related formulations of
> online regret-minimizing control that allow a low-regret adaptive controller to match (robust) $H\_\infty$
> control in the adversarial case, meaning that the controller becomes very conservative in adversarial
> settings. Further, [38] used a similar control-theoretic method on a nonlinear 6DOF drone model with
> learned $A$ and $B$ matrices (which were not exact), achieving good performance despite the attendant
> mismatch in disturbance specification. This class of algorithms tends to exhibit good robustness
> properties across a variety of uncertainties.
>
> References:
>
> [1] C. Jin, R. Ge, P. Netrapalli, S. M. Kakade, and M. I. Jordan. How to Escape Saddle Points
> Efficiently. In Proceedings of the 34th International Conference on Machine Learning, pages
> 1724–1732. PMLR, July 2017. URL https://proceedings.mlr.press/v70/jin17a.html.
> ISSN: 2640-3498.

---

> ### Author Response · Authors · 2023-08-11
> **Update and Additional Clarifications**
>
> Dear Reviewer, we would appreciate it if you could let us know if our rebuttal has addressed the points raised in your review. We are happy to engage in further discussions. Thank you again for your time and effort in providing feedback on the paper.

---

> > ### Author Response · Authors · 2023-08-14
> > **Follow-Up for Additional Clarifications**
> >
> > Dear Reviewer, we would appreciate it if you could let us know if our responses have addressed the points raised in your review. We are happy to engage in further discussions. Thank you again for your time and effort in providing feedback on the paper.

---

### Decision · Program_Chairs · 2023-08-30

**Decision:**

Accept (Poster)

**Comment:**

Scores:    DrfS: Weak Accept   x84o: Weak Reject,   63fg: Weak Accept

Quality: The paper presents an online learning algorithm for obstacle avoidance in robot navigation. The resulting policy is robust against uncertainties and perturbations and  in hindsight is provably comparable to the best obstacle avoidance policy (from a restricted class). Theoretical results are provided and extensive simulation and experimental results are conducted. All reviewers  have acknowledged that the paper presents a strong algorithm and is well executed.

Clarity: The paper is clear and well organized.

Originality: Good. The paper makes a novel technical contributions by formulating  the obstacle avoidance problem in a regret minimization framework and by developing a trust region-based online learning algorithm with provable bounds, which  is important for safety considerations.

Significance: As online obstacle avoidance algorithm with guaranteed safety bounds are rare, this paper provides a significant contribution.

Cons:

- the significance of the provided method in comparsion with other more data-driven, obstacle avoidance algorithm is not entirely clear

Pros:

- adaptive online obstacle avoidance algorithm (including dynamic obstacles) with policy safety certification (regret minimization)
- algorithm is robust against uncertainties and perturbations
- theoretical results are provided and extensive simulations and hardware experiments are conducted